# Synthesis of Four Orthogonally Protected Rare l-Hexose Thioglycosides from d-Mannose by C-5 and C-4 Epimerization

**DOI:** 10.3390/molecules27113422

**Published:** 2022-05-25

**Authors:** Fruzsina Demeter, Ilona Bereczki, Anikó Borbás, Mihály Herczeg

**Affiliations:** 1Department of Pharmaceutical Chemistry, University of Debrecen, Egyetem tér 1, H-4032 Debrecen, Hungary; demeter.fruzsina@science.unideb.hu (F.D.); bereczki.ilona@pharm.unideb.hu (I.B.); 2Research Group for Oligosaccharide Chemistry of Hungarian Academy of Sciences, ELKH, Egyetem tér 1, H-4032 Debrecen, Hungary

**Keywords:** l-hexoses, thioglycosides, epimerization, elimination, hydroboration

## Abstract

l-Hexoses are important components of biologically relevant compounds and precursors of some therapeuticals. However, they typically cannot be obtained from natural sources and due to the complexity of their synthesis, their commercially available derivatives are also very expensive. Starting from one of the cheapest d-hexoses, d-mannose, using inexpensive and readily available chemicals, we developed a reaction pathway to obtain two orthogonally protected l-hexose thioglycoside derivatives, l-gulose and l-galactose, through the corresponding 5,6-unsaturated thioglycosides by C-5 epimerization. From these derivatives, the orthogonally protected thioglycosides of further two l-hexoses (l-allose and l-glucose) were synthesized by C-4 epimerization. The preparation of the key intermediates, the 5,6-unsaturated derivatives, was systematically studied using various protecting groups. By the method developed, we are able to produce highly functionalized l-gulose derivatives in 9 steps (total yields: 21–23%) and l-galactose derivatives in 12 steps (total yields: 6–8%) starting from d-mannose.

## 1. Introduction

l-Sugars are key constituents of many biorelevant natural products such as polysaccharides, glycopeptides, and terpene or steroid glycosides (Figure 1) [1]. For example, l-guluronic acid is found in alginates (**I**) [2,3,4,5,6], the cell walls polysaccharides of brown algae, and l-gulose is a component of the potent antitumor antibiotic Bleomycin A2 (**II**) produced by *Streptomyces verticillus* [7,8,9]. The tetracyclic triterpene Datiscoside C (**III**), which is isolated from *Datisca glomerata*, contains a 6-deoxy-l-alloside [10], while l-galactose can be found in the A side chain of the pectic polysaccharide rhamnogalacturonan II (**IV**) [11]. l-Glucose is a major component of littoralisone isolated from *Verbena littoralis*, which has a nerve growth factor potentiating activity [12,13]. Apoptolidine A (**V**) macrolide antibiotic, isolated from *Nocardiopsis* species and used to initiate selective apoptosis of tumor cells, contains a 6-deoxy-l-glucose [14,15,16]. Furthermore, l-iduronic acid is a key component of the important mammalian glycosaminoglycans heparin, heparan sulfate, and dermatan sulfate [17,18,19].

l-Hexoses are rare sugars, much less common in nature than their enantiomers, the d-hexoses, and are thus very expensive [20], which prevents the exploitation of their biological potential. Therefore, it is important to produce l-sugars from common carbohydrates by cost-effective and high-yielding synthetic routes. To meet the demand of l-sugars, various strategies have been developed for their synthesis [20,21,22,23,24], including C-5 epimerization of readily available d-sugars [25,26,27,28,29,30,31,32,33,34,35,36,37], homologation of carbohydrates with shorter chains [38,39,40,41,42], de novo synthetic routes [43,44,45,46,47,48,49,50], head to tail inversion [51,52,53,54,55,56], site selective epimerization [57,58], C-H activation [59,60], and enzymatic synthesis [61,62]. However, there are very few approaches that provide the l-sugars as glycosyl donors, e.g., in form of thioglycosides, ready for glycosylation [60,63,64,65,66]. Thus, it is highly desirable to develop new methods to produce rare sugars directly in the form of functionalized glycosyl donors suitable for oligosaccharide synthesis.

Herein, we present the synthesis of four l-sugars, the l-gulose, -allose, -galactose, and -glucose, as their highly functionalized thioglycosides from a cheaply accessible d-mannosyl thioglycoside, using the elimination-hydroboration-oxidation-based C-5 epimerization as the key transformation. The synthesis of l-sugars by hydroboration/oxidation of d-sugar-derived 5-enopyranosides has been extensively studied on α-*O*-glycosides, focusing mainly on d-glucosides, and to a lesser extent also on sugars of other configurations. (Figure 1A) [25,26,67]. Recently, we have successfully extended the scope of this method to thioglycosides derived from d-glucose, thus giving direct, rapid access to l-idose glycosyl donors. (Figure 1B) [65,66]. We demonstrated that the α-anomeric configuration is crucial and the bulky C-4 substituent is advantageous for the high l-*ido* selectivity and that, despite the sensitivity of sulfur to oxidation, the over-oxidation into sulfoxide is negligible [65,66]. On the basis of these results, we envisioned the expeditious synthesis of l-gulose and l-galactose donors starting from d-mannosyl and d-altrosyl thioglycosides, which are readily accessible from one of the cheapest sugars, d-mannose. Subsequent protecting group manipulation and stereoselective C-4 epimerization by either the Mitsunobu reaction or oxidation/stereoselective reduction were planned to obtain l-allose and l-glucose in the form of their thioglycosides (Figure 1C).

The most common anomeric leaving group, thiophenyl, was chosen as the aglycone, and the incorporation of ether and ester protecting groups (that are non-participating and participating groups) into the C-2 position was also devised to make the resulting l-thioglycosides suitable for the synthesis of either α- or β-glycosides. Our synthetic design requires 5,6-unsaturated pyranosides as key intermediates on which hydroboration/oxidation can be performed. A well-documented method for preparing the pyranosyl exocyclic alkene is the base-mediated elimination of the corresponding 6-deoxy-6-iodo-glycosides, for which silver(I) fluoride (AgF) [68], potassium *tert*-butoxide (*t*-BuOK) [69], sodium hydride (NaH) [70], and 1,8-diazabicyclo(5.4.0)undec-7-ene (DBU) [71,72,73,74] reagents are the most commonly used. However, monitoring of this dehydroiodination step is notoriously difficult, and it often suffers from unwanted side reactions [65,66] that may be exacerbated in the presence of base-sensitive ester protecting groups. Therefore, special attention was paid to optimizing this elimination step to make the whole synthetic procedure reliable and efficient.

While the hydroboration of d-glucoside-derived enopyranosides has been studied extensively [25,26,75,76,77], only two publications have so far addressed the hydroboration of the corresponding mannosides [25,67]. The BH_3_·THF-mediated C-5 epimerization of 5-enomannoside **8** was reported to proceed with moderate diastereoselectivity [25], probably due to the steric hindrance by the C-2 substituent which hampers attack of the reagent from the β-face (Figure 1A). A much higher l-*gulo*:d-*manno* ratio could be achieved using catecholborane and RhCl(PPh_3_)_3_ catalysis (Wilkinson’s catalyst), but the protecting group pattern of mannoside alkenes also proved to be an important factor in l-*gulo* selectivity (**11**→**12**) [67]. As our goal was to produce highly functionalized, orthogonally protected thioglycoside donors via cost-effective routes, we focused on exploiting the stereoselectivity-enhancing effects of the different protecting groups and tried to avoid the use of expensive metal catalysts.

## 2. Results

First, the two key transformations, elimination and hydroboration/oxidation, were tested on the ether protected derivative **25**, which was obtained from the known phenyl 1-thio-α-d-mannoside derivative **23** [78,79] (Figure 2). Regioselective reductive 4,6-acetal opening of compound **23** gave the primary alcohol **24**, treatment of which with Ph_3_P, iodine, and imidazole afforded the 6-iodo-mannoside **25**. For the next dehydroiodination reaction, our choice was DBU as a commonly used elimination reagent, which is compatible with both ester and ether protecting groups.

Treatment of **25** with DBU (4 equiv.) in dry THF at reflux temperature (70 °C) for 5 h gave the expected unsaturated product **26**, however, only with moderate yield (43%) due to the incomplete conversion of **25** and formation of by-product **27** (Table 1). The latter sugar-amidinium salt, which results from a nucleophilic substitution of iodine **25** by DBU, is quite an unusual product. Bicyclic amidine DBU features poor nucleophilicity and strong basicity [80,81,82], and although there are an increasing number of examples of it reacting as a nucleophilic agent [83,84,85], this behavior has so far not been observed in HI elimination reactions. At the same time, similar 6-deoxy-6-ammonium salts of carbohydrates have already been prepared, which have been investigated as chiral ionic liquids [86] or antibacterial agents [87,88], however, they do not contain the DBU but the remarkably more nucleophilic DABCO (1,4-diazabicyclo[2.2.2]octane) at the primary position of pyranosides.

In order to improve the yield of **26**, the elimination reaction was carried out in dry toluene at elevated temperature (110 °C). After 2 h under reflux, higher conversion of the starting iodide was observed, however, the yield of **26** remained moderate (40%) and the ratio of **27** to **26** increased, indicating that the higher temperature favored the competing nucleophilic substitution rather than elimination. Changing back the solvent to THF and increasing the reaction time to 24 h yielded the expected 5,6-unsaturated derivative with 53% yield, but, unfortunately, the formation of the unwanted DBU-sugar conjugate **27** could not be suppressed.

The C-5 epimerization of **26** was performed by hydroboration with BH_3_·THF complex followed by oxidation with H_2_O_2_ under previously optimized conditions, using 10 equiv. of the borane reagent [25,26,65,66,67]. Although the yield of the expected l-*gulo*-configured product **28** was only 53%, a sufficiently high l-guloside: d-mannoside ratio was observed (**28**:**24**~5:1), so it was considered worthwhile to investigate the C-5 epimerization of 1-thiomannoside alkenes having other protecting group patterns.

### 2.1. Synthesis of l-Gulose and l-Allose Derivatives from d-Mannose

In order to thoroughly investigate the effect of ether and ester protecting groups on the production of l-hexoses, three different substitution patterns were installed on phenyl 1-thio-α-d-mannopyranoside. In each case, (2-naphthyl)methyl ether (NAP) was introduced to the C-4 position. The 2,3-hydroxyl groups, on the other hand, were masked in varying ways, using either di-*O*-benzyl or di-*O*-benzoyl or 2-*O*-benzoyl-3-*O*-benzyl protection (Figure 3). First, the 4,6-*O*-(2-naphthyl)methylene derivative of phenyl α-1-thio-d-mannoside (**29**) [89,90,91] was prepared from d-mannose, as a suitable starting material, in four steps using routine transformations. On route to the alternating ether–ester protecting group combination, the 3-*O*-benzyl-protected derivative **30** was formed by preparing the temporary 2,3-*O*-stannylene acetal derivative using dibutyltin oxide in dry toluene, followed by its reaction with BnBr and CsF. Position C-2 of **30** was esterified with BzCl in dry pyridine to give the expected 2-*O*-benzoyl-3-*O*-benzyl protected compound **31**. Alkylation of **29** with BnBr under basic conditions gave the 2,3-di-*O*-benzyl derivative **32**. The ester groups at positions 2 and 3 of diol **29** were formed with BzCl in dry pyridine to give **33**. In compounds **31**, **32**, and **33**, the primary hydroxyl group was liberated by regioselective reductive ring-opening reaction of the 4,6-acetal with BH_3_·THF/TMSOTf reagent combination in dry dichloromethane. The expected 6-OH derivatives **34**, **35**, and **36** were obtained in excellent yields with complete regioselectivity. Subsequently, the primary alcohols were converted to the 6-iodo derivatives **37**, **38**, and **39** by treatment with triphenylphosphine, iodine, and imidazole. Since the DBU-induced elimination did not work satisfactorily in the model reaction of **25**, the elimination reactions of **37**–**39** were studied using the three most common dehydrohalogenating reagents: NaH, DBU, and AgF.

The NaH induced elimination of the fully ether-protected mannoside **37** resulted in the expected 5,6-unsaturated **40** with an excellent, 91% yield (Figure 4).

Treatment of **37** with DBU led to, again, the simultaneous formation of the desired alkene derivative **40** and the 6-amidinium by-product **41** in a ~5:4 ratio. The HI elimination using AgF produced the exocyclic alkene derivative in high efficacy, however, a small amount of the corresponding 6-deoxy-6-fluoro derivative **42** was also obtained due to a concomitant nucleophilic substitution reaction on compound **37**.

The next tested compound was the 2,3-di-*O*-benzoyl derivative **38** (Figure 5). In the NaH-mediated reaction, instead of the fully protected exocyclic alkene, the unsaturated 2,3-diol **43** was formed since, as expected, the ester groups were cleaved under the used strongly basic conditions. Unfortunately, the yield of **43** was only 17%, and the 3,6-anhydro derivative **44** was isolated as the major product with 59% yield. The predominant formation of **44** in this reaction indicates that the ester-cleavage preceded the elimination reaction and the 6-iodo-2,3-diol intermediate formed rather suffered an intramolecular nucleophilic substitution by the 3-OH than an E2 elimination reaction by NaH.

The DBU-induced elimination reaction gave the expected 5,6-unsaturated compound **45** in good yield (66%). A nucleophilic substitution reaction by DBU was also observed, although to a slightly lesser extent than for the fully ether protected **25** and **37**, resulting in the quaternary amidinium salt **46** in 31% yield.

Dehydroiodination with AgF gave the expected compound **45** in moderate yield of 55%, and a double eliminated derivative (**47**) was isolated as the by-product. Formation of the 3-phenylthio glycal derivative **47** can be explained by elimination of the 3-OBz group followed by an allylic rearrangement reaction of the intermediate 2,3-unsaturated thioglycoside [92].

In the case of the 2-*O*-benzoyl-3-*O*-benzyl protected **39**, only the 2,6-anhydro derivative **48** was formed in the elimination with NaH, in excellent yield (99%) (Figure 6).

Using DBU as the eliminating agent, the expected unsaturated compound **49** was isolated in moderate yield (54%), and the amidinium by-product (**50**) was formed again in a competitive nucleophilic substitution reaction. For this derivative (**39**), the elimination reaction elicited by AgF gave the best result, the expected exocyclic alkene **49** was obtained in 87% yield and no by-product formation was observed.

After successful preparation of the 5,6-unsaturated derivatives, the C-5 epimerization reactions were performed on all three mannose-derived exocyclic alkenes (Figure 7). The epimerization process included hydroboration with BH_3_·THF complex in dry THF followed by oxidation with 30% H_2_O_2_, and hydrolysis of the resulting boronic acid ester under alkaline conditions with satd. NaHCO_3_ solution. The conversion of **40**, **45**, and **49** into l-series proceeded with acceptable-to-high stereoselectivity, producing the expected l-gulopyranosides **51**, **52**, and **53** with good to excellent yields. The protecting group patterns noticeably affected both the yield and stereoselectivity of the C-5 epimerization. While only a 5.4 to 1 l-*gulo*:d-*manno* ratio was obtained with the fully ether protected **40**, the l-*gulo* ratio significantly increased to 9.3:1 by changing the 2,3-benzyl groups into benzoyls in **45**.

The highest yield and the best l-*gulo* selectivity was achieved from **49** having the benzoyl group at position C-2. It is hypothesized that the C-2 ester group delivers the borane to the top face through coordination to the carbonyl group, and this promotes hydride donation from the upper side to the C-5 carbon. The d-*manno* derivatives (**34**, **35**, and **36**) were also isolated from the reaction mixtures which can be recycled and converted to the l-*gulo* configured product in three steps (iodination, elimination, C-5 epimerization).

The l-*gulo* derivatives (**51**, **52**, **53**) were converted to the corresponding l-allopyranosides by oxidation/reduction-based C-4 epimerization in four steps (Figure 8). First, the (2-naphthyl)methyl group was moved from position 4 to the primary position via a DDQ (2,3-dichloro-5,6-dicyanobenzoquinone) mediated oxidative acetal ring closure under strictly anhydrous conditions followed by a reductive acetal opening with BH_3_·Me_3_N/AlCl_3_ reagent combination of the obtaining **54**, **55**, and **56**.

The latter reaction gave the required 4-OH products **57**, **58**, and **59** in a regioselective way with excellent yields. Oxidation of the free hydroxyl groups using pyridinium chlorochromate (PCC) in dry CH_2_Cl_2_ provided the expected 4-keto derivatives **60**, **61**, and **62** in good yields. The final reductive transformation was performed with l-selectride, because reduction of ulosides with l-selectride [93] at low temperature has been reported to give *cis* vicinal diols with high selectivity [64,94]. Reduction of compounds **60** and **62** with l-selectride occurred, indeed, with high stereoselectivity producing the expected equatorial 4-OH-containing l-allopyranosides **63** and **65** in excellent yields. However, reduction of the 2,3-di-*O*-benzoyl derivative **61** resulted in an inseparable 5:1 mixture of the l-*gulo* (**58**) and l-*allo* (**64**) configured products. The epimeric ratio was determined on the basis of the ^1^H NMR spectrum showing the H-3 l-*gulo* signal at 5.75 ppm and H-3 l-*allo* signal at 6.10 ppm. The high *allo*-selectivity in the reduction of **63** and **65** can be explained by the Cram’s chelation model [95]. In the present case, the metal coordinates to the carbonyl and *O*-3, hindering the attack by the reagent from the top-face, thus the hydride attacks from the bottom-face resulting selectively in the *allo*-epimers. The low and opposite stereoselectivity observed in the reduction of **61** was surprising, but literature survey revealed similar examples when no chelate control was observed during the reduction in the presence of an adjacent ester group [64,96,97]. The probable cause is that an ester protected oxygen lacks chelating ability due to the electron-withdrawing properties of the ester group. In the case of **61**, coordination of the metal to *O*-6 might be possible at the bottom-side, facilitating the attack of the hydride from the upper-side.

### 2.2. Synthesis of l-Galactose and l-Glucose Derivatives from d-Mannose through d-altrose

After successful conversion of highly functionalized d-mannosides to l-gulopyranosyl and l-allopyranosyl donors, our attention turned to the synthesis of l-*galacto* and l-*gluco* derivatives, which is achievable from d-mannose through d-altrose. First, mannoside **29** was converted to the functionalized d-altrose thioglycosides **72** and **73** ready for the elimination reactions (Figure 9). The C-2 hydroxyl group of diol **29** was selectively protected with a benzoyl group via introduction of a cyclic orthobenzoate to the 2,3-*cis* diol followed by regioselective orthoester opening under acidic conditions to give the axial ester **66** in an acceptable yield. Oxidation-stereoselective reduction was applied for C-3 epimerization, taking advantage of the inability of the C-2 benzoyl group to form chelates thereby favouring the hydride attack from the sterically less crowded β-side and providing the required *trans* diol. Oxidation of the free hydroxyl of **66** with PCC followed by reduction of the keto derivative with NaBH_4_ indeed predominantly inverted the configuration of C-3 group to give the needed d-*altro* configured product **67** with 58% yield. This compound was converted to the 2,3-di-*O*-benzylated derivative **68** in two steps including Zemplén deacylation and benzylation of the liberated hydroxyls using NaH and BnBr. A dibenzoylated derivative was also formed from **67** by esterification with BzCl in dry pyridine to give the expected 2,3-di-*O*-benzoyl derivative **69** in good yield.

The C-6-OH group of the fully protected compounds (**68**, **69**) was liberated by regioselective reductive ring-opening reactions using the BH_3_·THF/TMSOTf reagent combination. The primary hydroxyl in the resulting **70** and **71** was converted to a good leaving group by the treatment of Ph_3_P, iodine, and imidazole to give the C-6-iodide derivatives **72** and **73** in excellent yields.

As in the case of mannosides, the dehydroiodination of altropyranosides **72** and **73** was studied using three different reagents: NaH, DBU, and AgF (Figure 10). In the elimination reaction of the fully ether protected **72** with NaH, the expected 5,6-unsaturated compound **74** was formed in excellent yield. This exocyclic alkene was also obtained in the DBU-induced reaction with acceptable yield (58%), however, the DBU-conjugated amidinium salt **75**, formed in a concomitant nucleophilic replacement reaction, was also isolated from the reaction mixture in 41% yield. The dehydrohalogenation by AgF proceeded much more slowly than in the case of the d-*manno* configured molecule. Even after a reaction time of 48 h, the unsaturated **74** was only obtained in moderate 52% yield. No by-product was isolated from this reaction, but 19% of the starting **72** was recovered.

Treatment of the 2,3-di-*O*-benzoylated derivative **73** with NaH led to, similarly to the d-mannose case, deacylation followed by dehydroiodination and concomitant formation of a 2,6-anhydro-altroside (**77**) by an intramolecular replacement of the 6-iodide (Figure 11). Here, in contrast to the *manno* case, the 5,6-unsaturated **76** was isolated as the major product with 43% yield, and the 2,6-anhydrosugar by-product **77** was formed only in 22% yield.

The diol **76** was efficiently converted to the required **78** by routine esterification with benzoyl chloride. In the DBU-induced reaction, formation of the expected alkene **78** was accompanied, as in the previous reactions, by a nucleophilic substitution reaction leading to the quaternary amidinium salt **79** (42%). The AgF mediated reaction yielded the expected 5,6-unsaturated **78** in a moderate yield (53%), and by-product **47**, derived from a secondary elimination of **78** to a 2,5-dienoside followed by an allylic rearrangement reaction, was also isolated. Importantly, this compound was identical to those formed from mannoside **38** under the same conditions.

After successfully synthesizing the 5,6-unsaturated **74** and **78**, C-5 epimerization was performed on both derivatives using the hydroboration/oxidation reaction described previously (Figure 12).

The expected l-*galacto* configured products were formed in the reactions in good yields (**80**, **81**) and the corresponding d-altrose epimers were not detected in the reaction mixtures. As observed for mannoside-derived alkenes, C-5 epimerization was more efficient with ester protection than with ether protection. Following the epimerization scheme developed for l-guloside, the obtained l-*galacto* derivatives (**80**, **81**) were converted to the corresponding l-*gluco* epimers (**86**, **87**) in three steps (Figure 13).

First, as before, in a combination of a DDQ-mediated oxidative acetal cyclization (**82**, **83**) and a regioselective reductive ring-opening reaction, the NAP group at position C-4 was moved to position C-6, thus releasing the 4-OH group (**84**, **85**). The 4,6-acetalated l-*galacto* derivative **83** was produced in crystalline form and its structure was confirmed by X-ray diffraction study (Figure 2).

In the presence of a chelating adjacent C-3-*O*-benzyl group to the resulting keto function after oxidation of **84**, the oxidation/reduction would result in reformation of the galactose isomer due to chelation to *O*-3 at the bottom face of the pyranose ring. Therefore, our attention turned to Mitsunobu isomerization as an appropriate method to convert the 3,4-*cis* diol of **84** and **85** to the required *trans* diol structure [64]. The Mitsunobu epimerization of C-4-OH was performed with *p*-nitrobenzoic acid, triphenylphosphine (Ph_3_P), and diisopropyl azodicarboxylate (DIAD) reagents in dry toluene. In the case of the ether-protected derivative, the fully protected product **86** with the desired l-*gluco* configuration was obtained in good yield. However, the 2,3-*O*-benzoyl-containing compound (**85**) could only be converted to the expected l-*gluco* configured product **87** in a moderate yield of 51%. We observed that an elimination reaction also took place producing the 4-deoxy-4,5-unsaturated derivative **88** as a by-product. This compound was formed by E2 elimination of the antiperiplanar 5-H_axial_ and the oxyphosphonium ion intermediate, which is presumably facilitated by the ester groups [64,98,99].

## 3. Materials and Methods

### 3.1. General Information

Optical rotations were measured at room temperature on a Perkin-Elmer 241 automatic polarimeter. TLC analysis was performed on Kieselgel 60 F_254_ (Merck Millipore, Burlington, MA, USA) silica-gel plates with visualization by immersing in a sulphuric-acid solution (5% in EtOH) followed by heating. Column chromatography was performed on silica gel 60 (Merck 0.063–0.200 mm). Organic solutions were dried over MgSO_4_ and concentrated under vacuum. ^1^H and J-modulated ^13^C NMR spectroscopy (^1^H: 400 and 500 MHz; ^13^C: 100.28 and 125.76 MHz) were performed on Bruker DRX-400 and Bruker Avance II 500 spectrometers at 25 °C. Chemical shifts are referenced to SiMe_4_ or sodium 3-(trimethylsilyl)-1-propanesulfonate (DSS, *δ* = 0.00 ppm for ^1^H nuclei) and to residual solvent signals (CDCl_3_: *δ* = 77.16 ppm, CD_3_OD: *δ* = 49.15 ppm for ^13^C nuclei). ESI-TOF MS spectra were recorded by a microTOF-Q type QqTOFMS mass spectrometer (Bruker) in the positive ion mode using MeOH as the solvent. HRMS measurements were carried out on a maXis II UHR ESI-QTOF MS instrument (Bruker) in positive ionization mode. The following parameters were applied for the electrospray ion source: capillary voltage: 3.6 kV; end plate offset: 500 V; nebulizer pressure: 0.5 bar; dry gas temperature: 200 °C; and dry gas flow rate: 4.0 L/min. Constant background correction was applied for each spectrum. The background was recorded before each sample by injecting the blank sample matrix (solvent). Na-formate calibrant was injected after each sample, which enabled internal calibration during data evaluation. Mass spectra were recorded by otofControl version 4.1 (build: 3.5, Bruker) and processed by Compass DataAnalysis version 4.4 (build: 200.55.2969). Deposition Number 2163274 for compound 83 contains the supplementary crystallo-graphic data for this paper (see Appendix A). These data are provided free of charge by the joint Cambridge Crystallographic Data Centre and Fachinformationszentrum Karlsruhe Access Structures service.

### 3.2. General Methods

#### 3.2.1. General Method A for Iodination (**25**, **37**, **38**, **39**, **72**, **73**)

To the solution of the corresponding 6-OH compound (**24**, **34**, **35**, **36**, **70**, and **71**) (2.662 mmol) in dry toluene (24 mL), triphenylphosphine (1.05 g, 3.993 mmol, 1.5 equiv.), imidazole (544 mg, 7.986 mmol, 3.0 equiv.), and iodine (946 mg, 3.727 mmol, 1.4 equiv.) were added. The reaction mixture was stirred at 75 °C for 30 min then cooled to room temperature. To the stirred mixture, NaHCO_3_ (1.0 g) in water (14 mL) was added at room temperature. After 5 min, 10% aqueous solution of Na_2_S_2_O_3_ (25 mL) was added and the mixture was diluted with EtOAc (250 mL) and washed with H_2_O (2 × 75 mL). The organic layer was separated, dried, filtered, and concentrated.

#### 3.2.2. General Method B for Elimination of Iodide with DBU (**26**, **40**, **45**, **49**, **74**, **78**)

To the solution of the appropriate 6-iodide compound (**25**, **37**, **38**, **39**, **72**, and **73**) (0.427 mmol), dry THF (10 mL) heated to 75 °C and DBU (254 μL, 1.708 mmol, 4.0 equiv.) was added. The reaction mixture was stirred at 75 °C for 24 h. When the TLC showed complete consumption of the starting material, the reaction mixture was concentrated under reduced pressure.

#### 3.2.3. General Method C for Hydroboration and Oxidation (**28**, **51**, **52**, **53**, **80**, **81**)

To the solution of the appropriate 5,6-unsaturated compound (**26**, **40**, **45**, **49**, **74**, and **78**) (0.969 mmol) in dry THF (2.5 mL), BH_3_·THF complex (1M in THF, 10.1 mL, 9.690 mmol, 10.0 equiv.) was added. The reaction mixture was stirred at 0 °C for 1.5 h. After 1.5 h, 30% aqueous solution of H_2_O_2_ (2.5 mL) and saturated aqueous solution of NaHCO_3_ (8.7 mL) were added. The reaction mixture was stirred at room temperature for 50 min (**28**) or 2 h (**51**, **52**, **53**, **80**, **81**). The mixture was diluted with EtOAc (150 mL) and washed with saturated aqueous solution of NH_4_Cl (2 × 25 mL), H_2_O (25 mL) and saturated aqueous solution of NaCl (25 mL) until neutral pH. The organic layer was dried and concentrated.

#### 3.2.4. General Method D for Stereoselective Ring Opening with BH_3_·THF/TMSOTf (**34**, **35**, **36**, **70**, **71**)

To a solution of the appropriate acetal (**31**, **32**, **33**, **68**, and **69**) (0.500 mmol) in dry CH_2_Cl_2_ (5.0 mL), BH_3_·THF complex (1M in THF, 2.5 mL, 2.500 mmol, 5.0 equiv.) and TMSOTf (14 μL, 0.075 mmol, 0.15 equiv.) were added at 0 °C and the reaction mixture was stirred under argon for 1.5 h (**34**), 2 h (**36**, **70**, **71**), or 2.5 h (**35**) at room temperature. Next, Et_3_N (1.0 mL) was added, followed by careful addition of MeOH until the H_2_ evolution ceased. The mixture was concentrated, and the residue was coevaporated with MeOH (3 × 10 mL).

#### 3.2.5. General Method E for Elimination of Iodide with NaH (**40**, **43**, **48**, **74**, **76**)

A vigorously stirred solution of the corresponding 6-iodide compound (**37**, **38**, **39**, **72**, and **73**) (0.703 mmol) in dry DMF (4.6 mL) was cooled to 0 °C, NaH (2.812 mmol, 4.0 equiv.) was added, and the reaction mixture was stirred at room temperature for 5 h (**43**), 6 h (**48**), or 24 h (**40**, **74**, **76**). After the complete disappearance of the starting material, MeOH (0.5 mL) was added, and the mixture was concentrated. The residue was dissolved in CH_2_Cl_2_ (50 mL) and washed with H_2_O (2 × 10 mL). The organic layer was separated, dried, filtered and concentrated.

#### 3.2.6. General Method F for Elimination of Iodide with AgF (**40**, **45**, **49**, **74**, **78**)

A solution of the appropriate 6-iodide compound (**37**, **38**, **39**, **72**, and **73**) (0.427 mmol) in dry pyridine (5.0 mL) under argon atmosphere AgF (270 mg, 2.135 mmol, 5.0 equiv.) was added and the mixture was stirred at room temperature for 24 h (**40**, **45**, **49**) or 48 h (**74**, **78**) in dark. After that time, the reaction mixture was diluted with Et_2_O (100 mL), and the solution was filtered through a pad of Cellite, washed with Et_2_O, and concentrated.

#### 3.2.7. General Method G for Ring Closing with DDQ (**54**, **55**, **56**, **82**, **83**)

To a vigorously stirred solution of the corresponding 6-OH compound (**51**, **52**, **53**, **80**, and **81**) (1.402 mmol) in dry CH_2_Cl_2_ (42 mL), DDQ (478 mg, 2.104 mmol, 1.5 equiv.) and 4 Å MS (400 mg) were added and the reaction mixture was stirred for 2 h at 0 °C. After 2 h, the mixture was diluted with CH_2_Cl_2_ (350 mL), filtered, washed with a saturated aqueous solution of NaHCO_3_ (2 × 65 mL) and H_2_O (2 × 65 mL). The organic layer was then dried, filtered and concentrated.

#### 3.2.8. General Method H for Stereoselective Ring Opening with Me_3_N·BH_3_/AlCl_3_ (**57**, **58**, **59**, **84**, **85**)

To a solution of the appropriate 4,6-acetal (**54, 55, 56, 82**, and **83**) (0.145 mmol) in dry THF (500 μL), 4 Å MS (111 mg) and Me_3_N·BH_3_ (64 mg, 0.874 mmol, 6.0 equiv.) were added and the reaction mixture was stirred for 30 min at room temperature. After 30 min, the reaction mixture was cooled to 0 °C and AlCl_3_ (117 mg, 0.874 mmol, 6.0 equiv.) was added, the cooling medium was removed, and the mixture was stirred at room temperature for 1 h. After 1 h, the reaction mixture was diluted with CH_2_Cl_2_ (100 mL) and washed with H_2_O (2 × 15 mL). The organic layer was dried, filtered, and concentrated.

#### 3.2.9. General Method I for Oxidation with PCC (**60**, **61**, **62**)

To the solution of the corresponding 4-OH compound (**57, 58**, and **59**) (0.270 mmol) in dry CH_2_Cl_2_ (3.5 mL), 4 Å MS (10 p.) and PCC (233 mg, 1.080 mmol, 4.0 equiv.) were added under argon and stirred for 5 h at room temperature. After 5 h, the reaction mixture was diluted with *n*-hexane/Et_2_O = 1:1 (5.0 mL) and filtered through a pad of Celite. The filtrate was concentrated under reduced pressure.

#### 3.2.10. General Method J for Reduction with l-Selectride (**63**, **64**, **65**)

l-Selectride (1.0 M in THF; 254 µL, 0.254 mmol, 1.5 equiv.) was added to a solution of the corresponding 4-keto compound (**60**, **61**, and **62**) (0.169 mmol) in dry THF (2.0 mL) at −78 °C under argon. The reaction mixture was stirred at –78 °C for 1 h, then the temperature was raised to −20 °C over a period of 1 h. Subsequently, the reaction was quenched with water, and the mixture was extracted with CH_2_Cl_2_ (3 × 25 mL). The combined organic extracts were washed with saturated aqueous solution of NaHCO_3_ and brine, dried, filtered, and concentrated.

#### 3.2.11. General Method K for C-4 Epimerization by Mitsunobu Reaction (**86**, **87**)

To the solution of the appropriate 4-OH compound (**84** and **85**) (0.040 mmol) in toluene (440 μL) Ph_3_P (0.242 mmol, 6.0 equiv.), *p*-nitrobenzoic acid (0.242 mmol, 6.0 equiv.) and diisopropyl azodicarboxylate (0.242 mmol, 6.0 equiv.) were added. The mixture was stirred at room temperature under argon atmosphere for 15 h. Subsequently, the reaction was quenched with saturated aqueous solution of NaHCO_3_, and the mixture was extracted with CH_2_Cl_2_ (3 × 25 mL). The combined organic extracts were washed with water (10 mL), dried, filtered, and concentrated.

#### 3.2.12. Synthesis of l-Gulose and l-Allose Derivatives from d-Mannose

*Phenyl 2,4-di-O-benzyl-3-O-(2′-naphthyl)methyl-1-thio-α-d-mannopyranoside* (**24**). The acetal derivative **23** [79] (2.53 g, 4.272 mmol) was dissolved in anhydrous CH_2_Cl_2_ (38.5 mL) and anhydrous Et_2_O (13 mL). LiAlH_4_ (729 mg, 19.224 mmol, 4.5 equiv.) was added, and then a solution of AlCl_3_ (2.57 g, 19.224 mmol, 4.5 equiv.) in anhydrous Et_2_O (13 mL) was added. The reaction mixture was stirred at 0 °C for 1 h. The reaction mixture was diluted with EtOAc (86 mL) and H_2_O (21.5 mL), the precipitated solid was filtered through a pad of Celite, and the filter cake was washed with ethyl acetate. The filtrate was washed with water (2 × 25 mL), dried over MgSO_4_, filtered and concentrated. The crude product was purified by silica gel chromatography (7:3 *n*-hexane/EtOAc) to give **24** (2.01 g, 79%) as a colorless syrup. [α]_D_^25^ +51.2 (*c* 0.54, CHCl_3_); *R*_f_ 0.38 (7:3 *n*-hexane/EtOAc); ^1^H NMR (500 MHz, CDCl_3_) *δ* = 7.84–7.26 (22H, arom.), 5.51 (s, 1H, H-1), 4.99 (d, *J* = 10.9 Hz, 1H, BnC*H*_2_a), 4.81–4.67 (m, 5H, BnC*H*_2_b, BnC*H*_2_, NAPC*H*_2_), 4.14–4.12 (m, 1H, H-5), 4.07 (t, *J* = 9.3 Hz, 1H, H-4), 4.02 (s, 1H, H-2), 3.96 (d, *J* = 8.9 Hz, 1H, H-3), 3.86–3.80 (m, 2H, H-6-a,b), 1.83 (t, *J* = 6.2 Hz, 1H, C-6-OH) ppm; ^13^C NMR (100 MHz, CDCl_3_) *δ* = 138.4, 137.9, 135.7, 134.0, 133.4, 133.1 (6C, 6 × C_q_ arom.), 132.0–126.0 (22C, arom.), 86.1 (1C, C-1), 80.2 (1C, C-3), 76.5 (1C, C-2), 75.5 (1C, NAP*C*H_2_), 74.9 (1C, C-4), 73.4 (1C, C-5), 72.4 (2C, 2 × Bn*C*H_2_), 62.3 (1C, C-6) ppm; ESI-TOF-MS: *m/z* calcd for C_37_H_36_NaO_5_S [M + Na]^+^ 615.2176; found: 615.2168.

*Phenyl 2,4-di-O-benzyl-6-deoxy-6-iodo-3-O-(2′-naphthyl)methyl-1-thio-α-d-mannopyranoside* (**25**). Compound **24** (1.94 g, 3.270 mmol) was converted to **25** according to general **Method A**. The crude product was purified by silica gel chromatography (8:2 *n*-hexane/EtOAc) to give **25** (1.52 g, 66%) as a colorless syrup. [α]_D_^25^ + 50.5 (*c* 0.21, CHCl_3_); *R*_f_ 0.72 (7:3 *n*-hexane/EtOAc); ^1^H NMR (500 MHz, CDCl_3_) *δ* = 7.84–7.22 (22H, arom.), 5.58 (s, 1H, H-1), 5.04 (d, *J* = 10.9 Hz, 1H, BnC*H*_2_a), 4.74–4.72 (m, 4H, NAPC*H*_2_, BnC*H*_2_), 4.64 (d, *J* = 12.3 Hz, 1H, BnC*H*_2_b), 4.04 (s, 1H, H-2), 3.96–3.93 (m, 3H, H-3, H-4, H-5), 3.53 (d, *J* = 10.3 Hz, 1H, H-6a), 3.43 (dd, *J* = 5.1 Hz, *J* = 10.4 Hz, 1H, H-6b) ppm; ^13^C NMR (125 MHz, CDCl_3_) *δ* = 138.4, 137.9, 135.5, 134.2, 133.4, 133.1 (6C, 6 × C_q_ arom.), 131.8–126.0 (22C, arom.), 85.9 (1C, C-1), 79.9, 78.7 (2C, C-3, C-4), 76.5 (1C, C-2), 75.7 (1C, NAP*C*H_2_), 72.2 (1C, C-5), 72.2, 72.1 (2C, 2 × Bn*C*H_2_), 7.1 (1C, C-6) ppm; ESI-TOF-MS: *m/z* calcd for C_37_H_35_INaO_4_S [M + Na]^+^ 725.1193; found: 725.1195.

*Phenyl 2,4-di-O-benzyl-3-O-(2′-naphthyl)methyl-1-thio-α-d-lyxo-hex-5-enopyranoside* (**26**) and *8-N-[phenyl 2,4-di-O-benzyl-6-deoxy-6-yl-3-O-(2′-naphthyl)methyl-1-thio-α-d-mannopyranoside]-1,8-diazabicyclo(5.4.0)undec-7-ene-iodide* (**27**).

**Reaction I.**: Compound **25** (439 mg, 0.625 mmol) was converted to **26** according to general **Method B** using DBU (372 µL, 2.500 mmol, 4.0 equiv.) and refluxed (70 °C) for 5 h. The crude product was purified by silica gel chromatography (8:2 *n*-hexane/acetone) to give **26** (155 mg, 43%) as a colorless syrup and **27** (45 mg, 12%) as a colorless syrup.

**Reaction II.**: To a solution of compound **25** (323 mg, 0.460 mmol) in dry toluene (4.0 mL), DBU (274 µL, 1.839 mmol, 4.0 equiv.) was added and the mixture was stirred for 2 h at 110 °C. The reaction mixture was diluted with CH_2_Cl_2_ (100 mL), washed with 10% aqueous solution of Na_2_S_2_O_3_ (20 mL), 1M aqueous solution of HCl (2 × 20 mL), saturated aqueous solution of NaHCO_3_ (20 mL), and H_2_O (2 × 20 mL) until neutral pH. The organic layer was dried over MgSO_4_, filtered, and concentrated. The crude product was purified by silica gel chromatography (8:2 *n*-hexane/acetone → 9:1 CH_2_Cl_2_/MeOH) to give **26** (107 mg, 40%) as a colorless syrup and **27** (73 mg, 22%) as a light yellow syrup.

**Reaction III.**: Compound **25** (328 mg, 0.467 mmol) was converted to **26** according to general **Method B** using DBU (278 µL, 1.868 mmol, 4.0 equiv.) and refluxed for 24 h. The crude product was purified by silica gel chromatography (8:2 *n*-hexane/acetone → 9:1 CH_2_Cl_2_/MeOH) to give **26** (141 mg, 53%) as a colorless syrup and **27** (94 mg, 28%) as a light yellow syrup.

Data of **26**: [α]_D_^25^ +53.5 (*c* 0.14, CHCl_3_); *R*_f_ 0.43 (8:2 *n*-hexane/acetone); ^1^H NMR (500 MHz, CDCl_3_) *δ* = 7.81–7.22 (m, 22H, arom), 5.48 (d, *J* = 4.7 Hz, 1H, H-1), 4.86–4.60 (m, 8H, H-6a,b, 2 × BnC*H*_2_, NAPC*H*_2_), 4.29 (d, *J* = 7.5 Hz, 1H, H-4), 4.02 (d, *J* = 3.3 Hz, 1H, H-2), 3.94 (dd, *J* = 1.6 Hz, *J* = 7.2 Hz, 1H, H-3) ppm; ^13^C NMR (125 MHz, CDCl_3_) *δ* = 154.5 (1C, C-5), 138.2, 137.9, 135.8, 133.7, 133.3, 133.1 (6C, 6 × C_q_ arom.), 132.0–125.9 (22C, arom.), 99.1 (1C, C-6), 86.4 (1C, C-1), 77.3 (1C, C-3), 76.3 (1C, C-4), 76.0 (1C, C-2), 73.1, 72.8 (3C, 2 × Bn*C*H_2_, NAP*C*H_2_) ppm; ESI-TOF-MS: *m/z* calcd for C_37_H_34_NaO_4_S [M + Na]^+^ 597.2070; found: 597.2070.

Data of **27**: [α]_D_^25^ +66.7 (*c* 0.24, CHCl_3_); *R*_f_ 0.49 (9:1 CH_2_Cl_2_/MeOH); ^1^H NMR (400 MHz, MeOD) *δ* = 7.85–7.26 (m, 22H, arom.), 5.83 (s, 1H, H-1), 5.00–4.63 (m, 6H, 2 × BnC*H*_2_, NAPC*H*_2_), 4.17–4.15 (m, 1H, H-2), 4.06 (td, *J* = 1.7 Hz, *J* = 9.4 Hz, 1H, H-5), 3.97 (dd, *J* = 3.0 Hz, *J* = 9.1 Hz, 1H, H-3), 3.75 (t, *J* = 9.5 Hz, 1H, H-4), 3.69–3.60 (m, 2H, H-6a,b), 3.54–3.46 (m, 2H, NC*H*_2_ DBU), 3.25–3.16 (m, 4H, 2 × NC*H*_2_ DBU), 2.70–2.59 (m, 2H, C*H*_2_ DBU), 1.67–1.27 (m, 8H, 4 × C*H*_2_ DBU) ppm; ^13^C NMR (100 MHz, MeOD) *δ* = 168.5 (1C, C_q_ DBU), 139.5, 139.3, 136.7, 134.7, 134.5, 134.1 (6C, 6 × C_q_ arom.), 132.3–127.2 (22C, arom.), 85.2 (1C, C-1), 80.9 (1C, C-3), 77.1, 76.9 (2C, C-2, C-4), 76.1, 73.8, 72.7 (3C, 2 × Bn*C*H_2_, NAP*C*H_2_), 72.3 (1C, C-5), 56.0, 55.7, 50.0, 49.6 (4C, C-6, 3 × N*C*H_2_ DBU), 29.2, 29.1, 26.8, 23.8, 20.7 (5C, 5 × *C*H_2_ DBU) ppm; UHR ESI-QTOF: *m/z* calcd for C_46_H_51_N_2_O_4_S [M]^+^ 727.3564; found: 727.3568.

*Phenyl 2,4-di-O-benzyl-3-O-(2′-naphthyl)methyl-1-thio-β-l-gulopyranoside* (**28**). Compound **26** (190 mg, 0.331 mmol) was converted to **28** according to general **Method C**. The crude product was purified by silica gel chromatography (7:3 *n*-hexane/EtOAc) to give **28** (104 mg, 53%) as a colorless syrup and **24** (24 mg, 12%) as a colorless syrup. [α]_D_^25^ +11.4 (*c* 0.21, CHCl_3_); *R*_f_ 0.33 (6:4 *n*-hexane/acetone); ^1^H NMR (500 MHz, CDCl_3_) *δ* = 7.82–6.94 (m, 22H, arom.), 5.27 (d, *J* = 9.8 Hz, 1H, H-1), 4.84–4.14 (m, 6H, 2 × BnC*H*_2_, NAP*C*H_2_), 3.98 (d, *J* = 4.5 Hz, 1H, H-5), 3.84–3.80 (m, 1H, H-6a), 3.76 (s, 2H, H-2, H-3), 3.50 (d, *J* = 8.4 Hz, 1H, H-6b), 3.39 (s, 1H, H-4), 1.91 (s, 1H, C-6-OH) ppm; ^13^C NMR (125 MHz, CDCl_3_) *δ* = 138.0, 137.6, 135.6, 134.1, 133.2, 133.1 (6C, 6 × C_q_ arom.), 131.7–126.1 (22C, arom.), 84.2 (1C, C-1), 76.0 (1C, C-5), 75.3 (1C, C-4), 75.1 (1C, C-3), 73.6, 73.1, 72.2 (3C, 2 × Bn*C*H_2_, NAP*C*H_2_), 72.7 (1C, C-2), 62.5 (1C, C-6) ppm; ESI-TOF-MS: *m/z* calcd for C_37_H_36_NaO_5_S [M + Na]^+^ 615.2176; found: 615.2174.

*Phenyl 3-O-benzyl-4,6-O-(2′-naphthyl)methylidene-1-thio-α-d-mannopyranoside* (**30**). To a solution of compound **29** [91] (4.09 g, 10.00 mmol) in dry toluene (71 mL), dibutyltin oxide (3.644 g, 15.00 mmol, 1.5 equiv.) was added. The solution was refluxed (110 °C) under argon atmosphere for 4 h. Then, after cooling to room temperature, the solvent was evaporated to dryness and the crude product was dried on high vacuum for 2 h. The residue was dissolved in dry DMF (36 mL), and CsF (3.04 g, 2.001 mmol) and BnBr (1.78 mL, 15.00 mmol, 1.5 equiv.) were added. The reaction mixture was stirred at 90 °C for 24 h. The reaction mixture was filtered and evaporated to dryness. The residue was dissolved in EtOAc (250 mL) and washed with H_2_O (2 × 25 mL). The organic layer was separated, dried over MgSO_4_, filtered, and concentrated under reduced pressure. The crude product was purified by crystallization from acetone/*n*-hexane to give **30** (3.59 g, 73%) as white crystals. [α]_D_^25^ +244.2 (*c* 0.19, CHCl_3_); M.p. = 194–197 °C (acetone/*n*-hexane); *R*_f_ 0.32 (7:3 *n*-hexane/acetone); ^1^H NMR (400 MHz, CDCl_3_) *δ* = 7.98–7.20 (m, 17H, arom.), 5.75 (s, 1H, H_ac_), 5.59 (s, 1H, H-1), 4.88 (d, *J* = 11.8 Hz, 1H, BnC*H*_2_a), 4.74 (d, *J* = 11.8 Hz, 1H, BnC*H*_2_b), 4.38 (td, *J* = 4.8 Hz, *J* = 9.7 Hz, 1H, H-5), 4.26–4.21 (m, 3H), 3.98 (dd, *J* = 3.2 Hz, *J* = 9.5 Hz, 1H), 3.90 (t, *J* = 10.2 Hz, 1H), 3.02 (d, *J* = 5.3 Hz, 1H, C-2-OH) ppm; ^13^C NMR (100 MHz, CDCl_3_) *δ* = 137.8, 134.9, 133.7, 133.4, 133.0 (5C, 5 × C_q_ arom.), 131.8–123.9 (17C, arom.), 101.9 (1C, C_ac_), 88.0 (1C, C-1), 79.2, 75.9, 71.5, 64.8 (4C, C-2, C-3, C-4, C-5), 73.3 (1C, Bn*C*H_2_), 68.7 (1C, C-6) ppm; ESI-TOF-MS: *m/z* calcd for C_30_H_28_NaO_5_S [M + Na]^+^ 523.1550; found: 523.1545.

*Phenyl 2-O-benzoyl-3-O-benzyl-4,6-O-(2′-naphthyl)methylidene-1-thio-α-d-mannopyranoside* (**31**). To a stirred solution of compound **30** (3.48 g, 5.807 mmol) in dry pyridine (15 mL), BzCl (0.85 mL, 7.259 mmol, 1.25 equiv./OH) was added at 0 °C and the reaction mixture was stirred for 24 h at room temperature. After 24 h, the mixture was diluted with CH_2_Cl_2_ (300 mL), washed with H_2_O (2 × 75 mL), 1M aqueous solution of H_2_SO_4_ (2 × 75 mL), H_2_O (2 × 75 mL), saturated aqueous solution of NaHCO_3_ (2 × 75 mL), and H_2_O (2 × 75 mL) until neutral pH. The organic layer was dried over MgSO_4_, filtered, and concentrated under reduced pressure. The crude product was purified by silica gel chromatography (7:3 *n*-hexane/acetone) to give **31** (3.56 g, 72%) as a colorless syrup. [α]_D_^25^ + 73.3 (*c* 0.34, CHCl_3_); *R*_f_ 0.52 (7:3 *n*-hexane/acetone); ^1^H NMR (500 MHz, CDCl_3_) *δ* = 8.12–7.20 (m, 22H, arom.), 5.87 (s, 1H, H_ac_), 5.85 (s, 1H, H-2), 5.64 (s, 1H, H-1), 4.76 (q, *J* = 12.2 Hz, 2H, BnC*H*_2_), 4.49–4.47 (m, 1H, H-5), 4.36–4.32 (m, 2H, H-4, H-6a), 4.17 (d, *J* = 9.6 Hz, 1H, H-3), 3.97 (t, *J* = 10.1 Hz, 1H, H-6b) ppm; ^13^C NMR (125 MHz, CDCl_3_) *δ* = 165.8 (1C, C_q_ Bz), 137.8, 134.9, 133.8, 133.2, 133.1, 129.7 (6C, 6 × C_q_ arom.), 133.5–123.9 (22C, arom.), 102.0 (1C, C_ac_), 87.4 (1C, C-1), 79.1 (1C, C-4), 74.4 (1C, C-3), 72.3 (1C, Bn*C*H_2_), 72.0 (1C, C-2), 68.7 (1C, C-6), 65.4 (1C, C-5) ppm; ESI-TOF-MS: *m/z* calcd for C_37_H_32_NaO_6_S [M + Na]^+^ 627.1812; found: 627.1819.

*Phenyl 2,3-di-O-benzyl-4,6-O-(2′-naphthyl)methylidene-1-thio-α-d-mannopyranoside* (**32**). To a solution of compound **29** [91] (3.99 g, 9.840 mmol) in dry DMF (43 mL), NaH (60%, 984 mg, 24.60 mmol, 1.25 equiv./OH) was added in portions at 0 °C. After 30 min at that temperature, BnBr (2.8 mL, 23.61 mmol, 1.25 equiv./OH) was added and the mixture was stirred for 24 h at room temperature. Next, MeOH (5.0 mL) was added, the reaction mixture was stirred for 15 min, and the solvents were evaporated. The residue was dissolved in CH_2_Cl_2_ (300 mL) and washed in H_2_O (3 × 50 mL) until neutral pH. The organic layer was dried over MgSO_4_ and concentrated. The crude product was purified by silica gel chromatography (9:1 *n*-hexane/acetone) to give **32** (3.75 g, 65%) as a colorless syrup. [α]_D_^25^ + 116.5 (*c* 0.36, CHCl_3_); *R*_f_ 0.29 (9:1 *n*-hexane/acetone); ^1^H NMR (400 MHz, CDCl_3_) *δ* = 8.00–7.22 (m, 22H, arom.), 5.80 (s, 1H, H_ac_), 5.53 (d, *J* = 1.2 Hz, 1H, H-1), 4.83 (d, *J* = 12.2 Hz, 1H, BnC*H*_2_a), 4.73 (s, 2H, BnC*H*_2_), 4.67 (d, *J* = 12.2 Hz, 1H, BnC*H*_2_b), 4.41–4.33 (m, 2H, H-5, H-6a), 4.27 (dd, *J* = 4.4 Hz, *J* = 10.2 Hz, 1H, H-6b), 4.07 (dd, *J* = 1.3 Hz, *J* = 3.1 Hz, 1H, H-2), 4.00 (dd, *J* = 3.2 Hz, *J* = 9.4 Hz, 1H, H-3), 3.94 (t, *J* = 9.8 Hz, 1H, H-4) ppm; ^13^C NMR (100 MHz, CDCl_3_) *δ* = 138.5, 137.9, 135.1, 133.9, 133.7, 133.1 (6C, 6 × C_q_ arom.), 131.8–123.9 (22C, arom.), 101.8 (1C, C_ac_), 87.3 (1C, C-1), 79.3, 78.2, 76.4 (3C, C-2, C-3, C-4), 73.2, 73.1 (2C, 2 × Bn*C*H_2_), 68.7 (1C, C-6), 65.6 (1C, C-5) ppm; ESI-TOF-MS: *m/z* calcd for: C_37_H_34_NaO_5_S [M + Na]^+^ 613.2019; found: 613.2039.

*Phenyl 2,3-di-O-benzoyl-4,6-O-(2′-naphthyl)methylidene-1-thio-α-d-mannopyranoside* (**33**). To a stirred solution of compound **29** [91] (4.10 g, 10.10 mmol) in pyridine (25 mL), BzCl (2.92 mL, 25.20 mmol, 1.25 equiv./OH) was added at 0 °C and the reaction mixture was stirred for 24 h at room temperature. After 24 h, the mixture was diluted with CH_2_Cl_2_ (300 mL), washed with H_2_O (150 mL), 1M aqueous solution of H_2_SO_4_ (2 × 150 mL), H_2_O (150 mL), saturated aqueous solution of NaHCO_3_ (2 × 150 mL), and H_2_O (150 mL) until neutral pH. The organic layer was separated, dried over MgSO_4_, filtered, and concentrated under reduced pressure. The crude product was purified by silica gel chromatography (7:3 *n*-hexane/acetone) to give **33** (4.38 g, 71%) as a colorless syrup. [α]_D_^25^ − 15.7 (*c* 0.22, CHCl_3_); *R*_f_ 0.46 (7:3 *n*-hexane/acetone); ^1^H NMR (500 MHz, CDCl_3_) *δ* = 8.10–7.23 (m, 22H, arom.), 5.99 (dd, *J* = 1.3 Hz, *J* = 3.4 Hz, 1H, H-2), 5.86 (dd, *J* = 3.4 Hz, *J* = 10.3 Hz, 1H, H-3), 5.84 (s, 1H, H_ac_), 5.70 (d, *J* = 1.1 Hz, 1H, H-1), 4.69 (td, *J* = 4.8 Hz, *J* = 9.8 Hz, 1H, H-5), 4.48 (t, *J* = 9.9 Hz, 1H, H-6a), 4.38 (dd, *J* = 4.9 Hz, *J* = 10.4 Hz, 1H, H-4), 4.03 (t, *J* = 10.3 Hz, 1H, H-6b) ppm; ^13^C NMR (125 MHz, CDCl_3_) *δ* = 165.6, 165.4 (2C, 2 × C_q_ Bz), 134.5, 133.8, 133.0, 129.7, 129.6 (6C, 6 × C_q_ arom.), 133.8–123.9 (22C, arom.), 102.3 (1C, C_ac_), 87.2 (1C, C-1), 77.2, 72.6, 69.4, 65.5 (4C, C-2, C-3, C-4, C-5), 68.8 (1C, C-6) ppm; ESI-TOF-MS: *m/z* calcd for: C_37_H_30_NaO_7_S [M + Na]^+^ 641.1604; found: 641.1603.

*Phenyl 2,3-di-O-benzyl-4-O-(2′-naphthyl)methyl-1-thio-α-d-mannopyranoside* (**34**). Compound **32** (3.46 g, 5.850 mmol) was converted to **34** according to general **Method D**. The crude product was purified by silica gel chromatography (7:3 *n*-hexane/acetone) to give **34** (3.44 g, 99%) as a colorless syrup. [α]_D_^25^ + 81.4 (*c* 0.37, CHCl_3_); *R*_f_ 0.39 (7:3 *n*-hexane/acetone); ^1^H NMR (500 MHz, CDCl_3_) *δ* = 7.81–7.27 (m, 22H, arom.), 5.52 (s, 1H, H-1), 5.10 (d, *J* = 11.1 Hz, 1H, BnC*H*_2_a), 4.83 (d, *J* = 11.0 Hz, 1H, BnC*H*_2_b), 4.69 (s, 2H, NAPC*H*_2_), 4.65 (s, 2H, BnC*H*_2_), 4.17–4.15 (m, 1H, H-5), 4.10 (t, *J* = 9.3 Hz, 1H, H-4), 4.02 (s, 1H, H-2), 3.92 (d, *J* = 8.9 Hz, 1H, H-3), 3.86–3.82 (m, 2H, H-6-a,b), 1.88 (s, 1H, C-6-OH) ppm; ^13^C NMR (100 MHz, CDCl_3_) *δ* = 138.2, 137.9, 135.9, 134.0, 133.4, 133.1 (6C, 6 × C_q_ arom.), 132.0–126.0 (22C, arom.), 86.2 (1C, C-1), 80.2 (1C, C-3), 76.5 (1C, C-2), 75.4 (1C, NAP*C*H_2_), 74.5 (1C, C-4), 73.4 (1C, C-5), 72.5, 72.3 (2C, 2 × Bn*C*H_2_), 62.3 (1C, C-6) ppm; ESI-TOF-MS: *m/z* calcd for C_37_H_36_NaO_5_S [M + Na]^+^ 615.2176; found: 615.2196.

*Phenyl 2,3-di-O-benzoyl-4-O-(2′-naphthyl)methyl-1-thio-α-d-mannopyranoside* (**35**). Compound **33** (4.83 g, 7.788 mmol) was converted to **35** according to general **Method D**. The crude product was purified by silica gel chromatography (65:35 *n*-hexane/acetone) to give **35** (4.16 g, 95%) as a colorless syrup. [α]_D_^25^ +8.1 (*c* 0.32, CHCl_3_); *R*_f_ 0.46 (6:4 *n*-hexane/acetone); ^1^H NMR (500 MHz, CDCl_3_) *δ* = 8.01–7.24 (m, 22H, arom.), 5.86 (d, *J* = 1.3 Hz, 1H, H-2), 5.79 (dd, *J* = 3.2 Hz, *J* = 8.9 Hz, 1H, H-3), 5.65 (s, 1H, H-1), 4.89 (s, 2H, NAPC*H*_2_), 4.44–4.41 (m, 2H, H-4, H-5), 3.96 (s, 2H, H-6a,b), 1.92 (s, 1H, H-6-OH) ppm; ^13^C NMR (125 MHz, CDCl_3_) *δ* = 165.5, 165.4 (2C, 2 × C_q_ Bz), 135.1, 133.3, 133.2, 133.1, 129.6 (6C, 6 × C_q_ arom.), 133.6–126.1 (22C, arom.), 86.2 (1C, C-1), 75.3 (1C, NAP*C*H_2_), 73.3, 73.0 (2C, C-4, C-5), 72.7 (1C, C-3), 72.6 (1C, C-2), 61.8 (1C, C-6) ppm; ESI-TOF-MS: *m/z* calcd for C_37_H_32_NaO_7_S [M + Na]^+^ 643.1761; found: 643.1742.

*Phenyl 2-O-benzoyl-3-O-benzyl-4-O-(2′-naphthyl)methyl-1-thio-α-d-mannopyranoside* (**36**). Compound **31** (3.56 g, 5.899 mmol) was converted to **36** according to general **Method D**. The crude product was purified by silica gel chromatography (7:3 *n*-hexane/acetone) to give **36** (2.82 g, 79%) as a colorless syrup. [α]_D_^25^ + 68.6 (*c* 0.29, CHCl_3_); *R*_f_ 0.42 (7:3 *n*-hexane/acetone); ^1^H NMR (500 MHz, CDCl_3_) *δ* = 8.06–7.24 (m, 22H, arom.), 5.86 (s, 1H, H-2), 5.59 (d, *J* = 0.9 Hz, 1H, H-1), 5.11–4.62 (m, 4H, NAPC*H*_2_, BnC*H*_2_), 4.29–4.27 (m, 1H, H-5), 4.12–4.11 (m, 2H, H-3, H-4), 3.90–3.87 (m, 2H, H-6a,b), 1.82 (t, *J* = 6.6 Hz, 1H, C-6-O*H*) ppm; ^13^C NMR (125 MHz, CDCl_3_) *δ* = 165.7 (1C, C_q_ Bz), 137.8, 135.7, 133.4, 133.2, 129.8 (6C, 6 × C_q_ arom.), 133.5–126.1 (22C, arom.), 86.6 (1C, C-1), 78.6 (1C, C-3), 75.5 (1C, NAP*C*H_2_), 74.2 (1C, C-4), 73.2 (1C, C-5), 71.9 (1C, Bn*C*H_2_), 70.9 (1C, C-2), 62.2 (1C, C-6) ppm; ESI-TOF-MS: *m/z* calcd for C_37_H_34_NaO_6_S [M + Na]^+^ 629.1968; found: 629.2028.

*Phenyl 2,3-di-O-benzyl-6-deoxy-6-iodo-4-O-(2′-naphthyl)methyl-1-thio-α-d-mannopyranoside* (**37**). Compound **34** (3.38 g, 5.702 mmol) was converted to **37** according to general **Method A**. The crude product was purified by silica gel chromatography (8:2 *n*-hexane/acetone) to give **37** (3.12 g, 78%) as a colorless syrup. [α]_D_^25^ +57.3 (*c* 0.63, CHCl_3_); *R*_f_ 0.42 (8:2 *n*-hexane/acetone); ^1^H NMR (500 MHz, CDCl_3_) *δ* = 7.82–7.22 (m, 22H, arom.), 5.59 (s, 1H, H-1), 5.15 (d, *J* = 11.1 Hz, 1H, BnC*H*_2_a), 4.88 (d, *J* = 11.1 Hz, 1H, BnC*H*_2_b), 4.73 (d, *J* = 12.3 Hz, 1H, BnC*H*_2_a), 4.65 (d, *J* = 12.2 Hz, 1H, BnC*H*_2_b), 4.61 (s, 2H, NAPC*H*_2_), 4.03 (s, 1H, H-2), 3.97–3.94 (m, 2H, H-4, H-5), 3.91–3.90 (m, 1H, H-3), 3.54 (d, *J* = 10.3 Hz, 1H, H-6a), 3.44 (dd, *J* = 5.4 Hz, *J* = 10.5 Hz, 1H, H-6b) ppm; ^13^C NMR (125 MHz, CDCl_3_) *δ* = 138.1, 137.9, 135.8, 134.3, 133.4, 133.2 (6C, 6 × C_q_ arom.), 131.7–126.1 (22C, arom.), 86.0 (1C, C-1), 79.9 (1C, C-3), 78.8 (1C, C-4), 76.4 (1C, C-2), 75.7 (1C, NAP*C*H_2_), 72.2 (1C, C-5), 72.1, 72.1 (2C, 2 × Bn*C*H_2_), 7.1 (1C, C-6) ppm; ESI-TOF-MS: *m/z* calcd for C_37_H_35_INaO_4_S [M + Na]^+^ 725.1193; found: 725.1183.

*Phenyl 2,3-di-O-benzoyl-6-deoxy-6-iodo-4-O-(2′-naphthyl)methyl-1-thio-α-d-mannopyranoside* (**38**). Compound **35** (4.08 g, 5.588 mmol) was converted to **38** according to general **Method A**. The crude product was purified by silica gel chromatography (7:3 *n*-hexane/acetone) to give **38** (3.77 g, 79%) as a colorless syrup. [α]_D_^25^ + 24.1 (*c* 0.22, CHCl_3_); *R*_f_ 0.53 (7:3 *n*-hexane/acetone); ^1^H NMR (400 MHz, CDCl_3_) *δ* = 8.13–7.23 (m, 22H, arom.), 5.92 (s, 1H), 5.80 (dd, *J* = 2.6 Hz, *J* = 9.5 Hz, 1H), 5.68 (s, 1H), 4.95 (s, 2H, NAPC*H*_2_), 4.32 (d, *J* = 9.4 Hz, 1H), 4.14–4.09 (m, 1H), 3.69 (dd, *J* = 3.8 Hz, *J* = 10.9 Hz, 1H, H-6a), 3.59–3.57 (m, 1H, H-6b) ppm; ^13^C NMR (100 MHz, CDCl_3_) *δ* = 165.4, 165.3 (2C, 2 × C_q_ Bz), 134.9, 133.2, 133.1, 133.0, 129.5, 129.4 (6C, 6 × C_q_ arom.), 133.6–125.9 (22C, arom.), 86.1 (1C, C-1), 75.7 (1C, NAP*C*H_2_), 77.7, 72.5, 72.3, 70.9 (4C, C-2, C-3, C-4, C-5), 8.6 (1C, C-6) ppm; ESI-TOF-MS: *m/z* calcd for C_37_H_31_INaO_6_S [M + Na]^+^ 753.0778; found: 753.0758.

*Phenyl 2-O-benzoyl-3-O-benzyl-6-deoxy-6-iodo-4-O-(2′-naphthyl)methyl-1-thio-α-d-mannopyranoside* (**39**). Compound **36** (2.77 g, 3.868 mmol) was converted to **39** according to general **Method A**. The crude product was purified by silica gel chromatography (8:2 *n*-hexane/acetone) to give **39** (3.23 g, 99%) as a colorless syrup. [α]_D_^25^ + 0.6 (*c* 0.14, CHCl_3_); *R*_f_ 0.43 (8:2 *n*-hexane/acetone); ^1^H NMR (500 MHz, CDCl_3_) *δ* = 8.16–7.24 (m, 22H, arom.), 5.90 (s, 1H, H-2), 5.62 (s, 1H, H-1), 5.15–4.60 (m, 4H, NAPC*H*_2_, BnC*H*_2_), 4.11 (d, *J* = 8.9 Hz, 1H, H-3), 4.02 (t, *J* = 9.1 Hz, 1H, H-4), 3.95 (d, *J* = 8.0 Hz, 1H, H-5), 3.61 (dd, *J* = 4.2 Hz, *J* = 10.8 Hz, 1H, H-6a), 3.54 (dd, *J* = 2.1 Hz, *J* = 10.7 Hz, 1H, H-6b) ppm; ^13^C NMR (125 MHz, CDCl_3_) *δ* = 165.7 (1C, C_q_ Bz), 137.6, 135.6, 133.5, 133.2, 129.8 (6C, 6 × C_q_ arom.), 133.5–126.1 (22C, arom.), 86.5 (1C, C-1), 78.4 (1C, C-4), 78.3 (1C, C-3), 75.8 (1C, NAP*C*H_2_), 71.8 (1C, Bn*C*H_2_), 71.1 (1C, C-5), 70.6 (1C, C-2), 8.5 (1C, C-6) ppm; ESI-TOF-MS: *m/z* calcd for C_37_H_33_INaO_5_S [M + Na]^+^ 739.0985; found: 739.0992.

*Phenyl 2,3-di-O-benzyl-4-O-(2′-naphthyl)methyl-1-thio-α-d-lyxo-hex-5-enopyranoside* (**40**). Compound **37** (300 mg, 0.427 mmol) was converted to **40** according to general **Method E**. The crude product was purified by silica gel chromatography (8:2 *n*-hexane/acetone) to give **40** (224 mg, 91%) as a colorless syrup. [α]_D_^25^ + 62.6 (*c* 0.19, CHCl_3_); *R*_f_ 0.38 (8:2 *n*-hexane/acetone); ^1^H NMR (400 MHz, CDCl_3_) *δ* = 7.79–7.18 (m, 22H, arom.), 5.51 (d, *J* = 4.7 Hz, 1H, H-1), 4.86–4.57 (m, 8H, H-6a,b, 2 × BnC*H*_2_, NAPC*H*_2_), 4.32 (d, *J* = 7.6 Hz, 1H, H-4), 4.04 (t, *J* = 3.5 Hz, 1H, H-2), 3.92 (dd, *J* = 2.1 Hz, *J* = 7.6 Hz, 1H, H-3) ppm; ^13^C NMR (100 MHz, CDCl_3_) *δ* = 154.4 (1C, C-5), 138.2, 137.8, 135.6, 133.7, 133.3, 133.0 (6C, 6 × C_q_ arom.), 131.8–125.8 (22C, arom.), 99.1 (1C, C-6), 86.3 (1C, C-1), 77.3 (1C, C-3), 76.1 (1C, C-4), 75.8 (1C, C-2), 72.8, 72.7, 72.6 (3C, 2 × Bn*C*H_2_, NAP*C*H_2_) ppm; ESI-TOF-MS: *m/z* calcd for C_37_H_34_NaO_4_S [M + Na]^+^ 597.2070; found: 597.2058.

*Phenyl 2,3-di-O-benzyl-4-O-(2′-naphthyl)methyl-1-thio-α-d-lyxo-hex-5-enopyranoside* (**40**) and *8-N-[phenyl 2,3-di-O-benzyl-6-deoxy-6-yl-4-O-(2′-naphthyl)methyl-1-thio-α-d-mannopyranoside]-1,8-diazabicyclo(5.4.0)undec-7-ene-iodide* (**41**). Compound **37** (300 mg, 0.427 mmol) was converted to **40** according to general **Method B**. The crude product was purified by silica gel chromatography (7:3 CH_2_Cl_2_/n-hexane → 9:1 CH_2_Cl_2_/MeOH) to give **40** (117 mg, 48%) as a colorless syrup and **41** (175 mg, 38%) as a light yellow syrup.

Data of **41**: [α]_D_^25^ + 67.7 (*c* 0.30, CHCl_3_); *R*_f_ 0.49 (9:1 CH_2_Cl_2_/MeOH); ^1^H NMR (500 MHz, CDCl_3_) *δ* = 7.86–7.27 (m, 22H, arom.), 5.60 (s, 1H, H-1), 5.11–4.61 (m, 6H, 2 × BnC*H*_2_, NAPC*H*_2_), 4.07 (t, *J* = 9.3 Hz, 1H, H-5), 3.97 (s, 1H, H-2), 3.92–3.87 (m, 2H, H-3, H-6a), 3.77 (t, *J* = 9.3 Hz, 1H, H-4), 3.63–3.53 (m, 3H, H-6b, NC*H*_2_ DBU), 3.46–3.40 (m, 4H, 2 × NC*H*_2_ DBU), 2.77–2.63 (m, 2H, C*H*_2_ DBU), 1.77–1.43 (m, 8H, 4 × C*H*_2_ DBU) ppm; ^13^C NMR (125 MHz, CDCl_3_) *δ* = 167.1 (1C, C_q_ DBU), 137.5, 134.9, 133.2, 133.0, 132.7 (6C, 6 × C_q_ arom.), 130.8–126.1 (22C, arom.), 84.5 (1C, C-1), 79.8 (1C, C-3), 75.6 (2C, C-2, C-4), 75.4, 72.7, 72.1 (3C, 2 × Bn*C*H_2_, NAP*C*H_2_), 55.7 (1C, N*C*H_2_ DBU), 55.0 (1C, C-6), 49.4, 48.2 (2C, 2 × N*C*H_2_ DBU), 28.8, 28.2, 25.7, 22.5, 19.7 (5C, 5 × *C*H_2_ DBU) ppm; UHR ESI-QTOF: *m/z* calcd for C_46_H_51_N_2_O_4_S [M]^+^ 727.3564; found: 727.3562.

*Phenyl 2,3-di-O-benzyl-4-O-(2′-naphthyl)methyl-1-thio-α-d-lyxo-hex-5-enopyranoside* (**40**) and *Phenyl 2,3-di-O-benzyl-6-deoxy-6-fluoro-4-O-(2′-naphthyl)methyl-1-thio-α-d-mannopyranoside* (**42**). Compound **37** (300 mg, 0.427 mmol) was converted to **40** according to general **Method F**. The crude product was purified by silica gel chromatography (7:3 CH_2_Cl_2_/n-hexane) to give **40** (210 mg, 86%) as a colorless syrup and **42** (14 mg, 6%) as a colorless syrup.

Data of **42**: [α]_D_^25^ + 60.6 (*c* 1.44, CHCl_3_); *R*_f_ 0.33 (7:3 CH_2_Cl_2_/*n*-hexane); ^1^H NMR (500 MHz, CDCl_3_) *δ* = 7.84–7.25 (m, 22H, arom.), 5.59 (d, *J* = 1.3 Hz, 1H, H-1), 5.14–4.54 (m, 8H, H-6a,b, 2 × BnC*H*_2_, NAPC*H*_2_), 4.26 (ddd, *J* = 2.7 Hz, *J* = 10.0 Hz, *J* = 27.4 Hz, 1H, H-5), 4.12 (t, *J* = 9.6 Hz, 1H, H-4), 4.03 (dd, *J* = 1.8 Hz, *J* = 2.9 Hz, 1H, H-2), 3.91 (dd, *J* = 3.0 Hz, *J* = 9.3 Hz, 1H, H-3) ppm; ^13^C NMR (125 MHz, CDCl_3_) *δ* = 138.1, 137.8, 135.8, 134.3, 133.4, 133.1 (6C, 6 × C_q_ arom.), 131.4–126.1 (22C, arom.), 86.0 (1C, C-1), 82.3 (1C, *J*_C,F_ = 172.9 Hz, H-6), 80.2, 76.2 (2C, C-2, C-3), 75.2 (1C, NAP*C*H_2_), 74.9 (1C, *J*_C,F_ = 6.6 Hz, C-5), 72.4 (1C, C-4), 72.2 (2C, 2 × Bn*C*H_2_) ppm; ESI-TOF-MS: *m/z* calcd for C_37_H_35_FNaO_4_S [M + Na]^+^ 617.2132; found: 617.2123.

*Phenyl 4-O-(2′-naphthyl)methyl-1-thio-α-d-lyxo-hex-5-enopyranoside* (**43**) and *Phenyl 3,6-anhydro-4-O-(2′-naphthyl)methyl-1-thio-α-d-mannopyranoside* (**44**) Compound **38** (300 mg, 0.411 mmol) was converted to **43** according to general **Method E**. The crude product was purified by silica gel chromatography (1:1 n-hexane/EtOAc) to give **43** (21 mg, 17%) as a colorless syrup and **44** (95 mg, 59%) as a colorless syrup.

Data of **43**: [α]_D_^25^ +70.4 (*c* 0.28, CHCl_3_); *R*_f_ 0.42 (1:1 *n*-hexane/EtOAc); ^1^H NMR (500 MHz, CDCl_3_) *δ* = 7.84–7.26 (m, 12H, arom.), 5.29 (d, *J* = 6.2 Hz, 1H, H-1), 4.95 (s, 1H, H-6a), 4.89 (d, *J* = 11.9 Hz, 1H, NAPC*H*_2_a), 4.79 (s, 1H, H-6b), 4.59 (d, *J* = 11.9 Hz, 1H, NAPC*H*_2_b), 4.15–4.13 (m, 2H, H-2, H-3), 4.10 (d, *J* = 5.6 Hz, 1H, H-4), 2.53, 2.49 (2 × s, 2H, H-2-O*H*, H-3-O*H*) ppm; ^13^C NMR (125 MHz, CDCl_3_) *δ* = 153.8 (1C, C-5), 135.2, 133.4, 133.2, 132.5 (4C, 4 × C_q_ arom.), 132.3–125.8 (12C, arom.), 101.1 (1C, C-6), 87.6 (1C, C-1), 76.9 (1C, C-4), 71.5 (1C, NAP*C*H_2_), 70.3 (1C, C-3), 68.5 (1C, C-2) ppm; ESI-TOF-MS: *m/z* calcd for C_23_H_22_NaO_4_S [M + Na]^+^ 417.1131; found: 417.1136.

Data of **44**: [α]_D_^25^ + 40.7 (*c* 0.28, CHCl_3_); *R*_f_ 0.28 (1:1 *n*-hexane/EtOAc); ^1^H NMR (500 MHz, CDCl_3_) *δ* = 7.83–7.09 (m, 12H, arom.), 4.93 (d, *J* = 8.7 Hz, 1H, H-1), 4.78 (d, *J* = 11.9 Hz, 1H, NAPC*H*_2_a), 4.61 (d, *J* = 11.9 Hz, 1H, NAPC*H*_2_b), 4.47 (t, *J* = 2.8 Hz, 1H, H-5), 4.32 (dd, *J* = 0.8 Hz, *J* = 6.0 Hz, 1H, H-3), 4.12 (d, *J* = 10.9 Hz, 1H, H-6a), 3.99 (dd, *J* = 2.7 Hz, *J* = 6.0 Hz, 1H, H-4), 3.93 (dd, *J* = 2.7 Hz, *J* = 10.8 Hz, 2H, H-2, H-6b), 3.91 (t, *J* = 8.0 Hz, H-2), 2.46 (s, 1H, H-2-O*H*) ppm; ^13^C NMR (125 MHz, CDCl_3_) *δ* = 134.7, 133.2, 133.1, 132.7 (4C, 4 × C_q_ arom.), 132.5–125.7 (12C, arom.), 85.6 (1C, C-1), 77.4 (1C, C-4), 76.9 (1C, C-3), 74.2 (1C, C-5), 71.9 (1C, NAP*C*H_2_), 69.5 (1C, C-6), 67.6 (1C, C-2) ppm; ESI-TOF-MS: *m/z* calcd for C_23_H_22_NaO_4_S [M + Na]^+^ 417.1131; found: 417.1129.

*Phenyl 2,3-di-O-benzoyl-4-O-(2′-naphthyl)methyl-1-thio-α-d-lyxo-hex-5-enopyranoside* (**45**) and *8-N-[phenyl 2,3-di-O-benzoyl-6-deoxy-6-yl-4-O-(2′-naphthyl)methyl-1-thio-α-d-mannopyranoside]-1,8-diazabicyclo(5.4.0)undec-7-ene-iodide* (**46**). Compound **38** (2.3 g, 3.150 mmol) was converted to **45** according to general **Method B**. The crude product was purified by silica gel chromatography (7:3 CH_2_Cl_2_/n-hexane → 95:5 CH_2_Cl_2_/MeOH) to give **45** (1.25 g, 66%) as a colorless syrup and **46** (773 mg, 31%) as a light yellow syrup.

Data of **45**: [α]_D_^25^ − 13.7 (*c* 0.27, CHCl_3_); *R*_f_ 0.28 (7:3 CH_2_Cl_2_/*n*-hexane); ^1^H NMR (400 MHz, CDCl_3_) *δ* = 7.89–7.25 (m, 22H, arom.), 5.84–5.82 (m, 1H, H-2), 5.78 (dd, *J* = 3.2 Hz, *J* = 8.1 Hz, 1H, H-3), 5.63 (d, *J* = 4.6 Hz, 1H, H-1), 5.03 (s, 1H, H-6a), 5.00 (s, 1H, H-6b), 4.96 (d, *J* = 12.0 Hz, 1H, NAPC*H*_2_a), 4.75 (d, *J* = 12.0 Hz, 1H, NAPC*H*_2_b), 4.50 (d, *J* = 8.1 Hz, 1H, H-4) ppm; ^13^C NMR (100 MHz, CDCl_3_) *δ* = 165.2 (2C, 2 × C_q_ Bz), 153.4 (1C, C-5) 134.9, 133.3, 133.2, 132.5, 129.5, 129.4 (6C, 6 × C_q_ arom.), 133.5–126.1 (22C, arom.), 100.9 (1C, C-6), 86.1 (1C, C-1) 74.0 (1C, C-4), 72.6 (1C, NAP*C*H_2_), 70.6 (2C, C-2, C-3) ppm; ESI-TOF-MS: *m/z* calcd for C_37_H_30_NaO_6_S [M + Na]^+^ 625.1655; found: 625.1665.

Data of **46**: [α]_D_^25^ + 27.4 (*c* 0.19, CHCl_3_); *R*_f_ 0.31 (95:5 CH_2_Cl_2_/MeOH); ^1^H NMR (500 MHz) *δ* = 8.05–7.26 (m, 22H, arom.), 5.86 (dd, *J* = 1.6 Hz, *J* = 3.1 Hz, 1H, H-2), 5.81 (d, *J* = 1.1 Hz, 1H, H-1), 5.78 (dd, *J* = 3.2 Hz, *J* = 9.4 Hz, 1H, H-3), 4.90 (d, *J* = 3.2 Hz, 2H, NAPC*H*_2_), 4.42 (t, *J* = 9.3 Hz, 1H, H-5), 4.23 (d, *J* = 15.0 Hz, 1H, H-6a), 4.12 (t, *J* = 9.6 Hz, 1H, H-4), 3.97 (dd, *J* = 9.7 Hz, *J* = 15.8 Hz, 1H, H-6b), 3.68–3.41 (m, 6H, 3 × NC*H*_2_ DBU), 2.80–2.79 (m, 2H, C*H*_2_ DBU), 1.94–1.56 (m, 8H, 4 × C*H*_2_ DBU) ppm; ^13^C NMR (125 MHz, CDCl_3_) *δ* = 167.6 (1C, C_q_ DBU), 165.4, 165.3 (2C, 2 × C_q_ Bz), 134.2, 133.1, 132.3, 129.0, 128.9 (6C, 6 × C_q_ arom.), 133.9–126.1 (22C, arom.), 84.6 (1C, C-1), 75.9 (1C, NAP*C*H_2_), 75.0 (1C, C-4), 72.5 (1C, C-3), 72.1 (1C, C-5), 71.6 (1C, C-2), 55.8 (1C, N*C*H_2_ DBU), 55.6 (1C, C-6), 49.5, 48.5 (2C, 2 × N*C*H_2_ DBU), 29.3, 28.5, 25.8, 22.7, 19.9 (5C, 5 × *C*H_2_ DBU) ppm; UHR ESI-QTOF: *m/z* calcd for C_46_H_47_N_2_O_6_S [M]^+^ 755.3149; found: 755.3147.

*Phenyl 2,3-di-O-benzoyl-4-O-(2′-naphthyl)methyl-1-thio-α-d-lyxo-hex-5-enopyranoside* (**45**) and *1,5-anhydro-2-O-benzoyl-3,6-dideoxy-4-O-(2′-naphthyl)methyl-3-S-phenyl-3-thio-α-d-erythro-hex-1,5-dienitol* (**47**). Compound **38** (300 mg, 0.411 mmol) was converted to **45** according to general **Method F**. The crude product was purified by silica gel chromatography (7:3 CH_2_Cl_2_/n-hexane) to give **47** (24 mg, 9%) as a colorless syrup and **45** (154 mg, 55%) as a colorless syrup.

Data of **47**: [α]_D_^25^ −18.8 (*c* 0.25, CHCl_3_); *R*_f_ 0.60 (7:3 CH_2_Cl_2_/*n*-hexane); ^1^H NMR (400 MHz, CDCl_3_) *δ* = 7.94–7.27 (m, 17H, arom.), 6.54 (s, 1H, H-1), 5.19 (d, *J* = 6.3 Hz, 1H, H-4), 5.08 (s, 2H, NAPC*H*_2_), 5.00 (s, 1H, H-6a), 4.69 (s, 1H, H-6b), 4.09 (dd, *J* = 1.0 Hz, *J* = 6.3 Hz, 1H, H-3) ppm; ^13^C NMR (100 MHz, CDCl_3_) *δ* = 164.9 (1C, C_q_ Bz), 148.5 (1C, C-5), 147.0 (1C, C-2), 133.8, 133.4, 133.2, 133.0, 129.5 (5C, 5 × C_q_ arom.), 133.6–125.1 (17C, arom.), 94.1 (1C, C-6), 93.6 (1C, C-4), 92.8 (1C, C-1), 69.7 (1C, NAP*C*H_2_), 46.4 (1C, C-3) ppm; ESI-TOF-MS: *m/z* calcd for C_30_H_24_NaO_4_S [M + Na]^+^ 503.1288; found: 503.1281.

*Phenyl 2,6-anhydro-3-O-benzyl-4-O-(2′-naphthyl)methyl-1-thio-α-d-mannopyranoside* (**48**). Compound **39** (310 mg, 0.433 mmol) was converted to **48** according to general **Method E**. The crude product was purified by silica gel chromatography (7:3 *n*-hexane/EtOAc) to give **48** (209 mg, 99%) as a colorless syrup. [α]_D_^25^ + 64.7 (*c* 0.15, CHCl_3_); *R*_f_ 0.37 (7:3 *n*-hexane/EtOAc); ^1^H NMR (500 MHz, CDCl_3_) *δ* = 8.09–7.20 (m, 17H, arom.), 5.68 (s, 1H, H-1), 4.81–4.52 (m, 4H, NAPC*H*_2_, BnC*H*_2_), 4.20–4.12 (m, 4H, H-2, H-3, H-5, H-6a), 3.813 (d, *J* = 9.8 Hz, 1H, H-6b), 3.73 (s, 1H, H-4) ppm; ^13^C NMR (125 MHz, CDCl_3_) *δ* = 137.6, 135.3, 134.3, 133.3, 133.1 (5C, 5 × C_q_ arom.), 133.7–126.0 (17C, arom.), 86.9 (1C, C-1), 80.6 (1C, C-4), 78.9 (1C, C-3), 71.0, 70.6 (2C, NAP*C*H_2_, Bn*C*H_2_), 69.9 (1C, C-2), 69.7 (1C, C-5), 66.8 (1C, C-6) ppm; ESI-TOF-MS: *m/z* calcd for C_30_H_28_NaO_4_S [M + Na]^+^ 507.1601; found: 507.1626.

*Phenyl 2-O-benzoyl-3-O-benzyl-4-O-(2′-naphthyl)methyl-1-thio-α-d-lyxo-hex-5-enopyranoside* (**49**) and *8-N-[phenyl 2-O-benzoyl-3-O-benzyl-6-deoxy-6-yl-4-O-(2′-naphthyl)methyl-1-thio-α-d-mannopyranoside]-1,8-diazabicyclo(5.4.0)undec-7-ene-iodide* (**50**).

**Reaction I.**: Compound **39** (336 mg, 0.469 mmol) was converted to **49** according to general **Method B**. The crude product was purified by silica gel chromatography (7:3 CH_2_Cl_2_/*n*-hexane → 95:5 CH_2_Cl_2_/MeOH) to give **49** (149 mg, 54%) as a colorless syrup and **50** (118 mg, 29%) as a light yellow syrup.

**Reaction II.**: Compound **39** (1.388 g, 1.938 mmol) was converted to **49** according to general **Method F**. The crude product was purified by silica gel chromatography (7:3 *n*-hexane/EtOAc) to give **49** (991 mg, 87%) as a colorless syrup.

Data of **49**: [α]_D_^25^ + 40.9 (*c* 0.35, CHCl_3_); *R*_f_ 0.60 (7:3 CH_2_Cl_2_/*n*-hexane); ^1^H NMR (400 MHz, CDCl_3_) *δ* = 8.03–7.19 (m, 22H, arom.), 5.80 (d, *J* = 3.5 Hz, 1H, H-2), 5.62 (d, *J* = 4.1 Hz, H-1), 5.00 (s, 1H, H-6a), 4.94–4.62 (m, 5H, H-6b, NAPC*H*_2_, BnC*H*_2_), 4.35 (d, *J* = 8.1 Hz, 1H, H-4), 4.12 (dd, *J* = 2.9 Hz, *J* = 8.1 Hz, 1H, H-3) ppm; ^13^C NMR (100 MHz, CDCl_3_) *δ* = 165.5 (1C, C_q_ Bz), 154.2 (1C, C-5), 137.7, 135.4, 133.4, 133.1, 133.0, 129.7 (6C, 6 × C_q_ arom.), 133.4–126.0 (22C, arom.), 99.8 (1C, C-6), 86.3 (1C, C-1), 76.8 (1C, C-3), 75.5 (1C, C-4), 73.2, 72.5 (2C, NAP*C*H_2_, Bn*C*H_2_), 70.6 (1C, C-2) ppm; ESI-TOF-MS: *m/z* calcd for C_37_H_32_NaO_5_S [M + Na]^+^ 611.1863; found: 611.1897.

Data of **50**: [α]_D_^25^ + 67.3 (*c* 0.26, CHCl_3_); *R*_f_ 0.63 (9:1 CH_2_Cl_2_/MeOH); ^1^H NMR (400 MHz) *δ* = 8.08–7.25 (m, 22H, arom.), 5.82 (dd, *J* = 1.5 Hz, *J* = 2.7 Hz, 1H, H-2), 5.73 (s, 1H, H-1), 5.10–4.59 (m, 4H, NAPC*H*_2_, BnC*H*_2_), 4.23 (t, *J* = 9.4 Hz, 1H, H-5), 4.10 (dd, *J* = 3.1 Hz, *J* = 9.0 Hz, 1H, H-3), 4.04 (d, *J* = 15.0 Hz, 1H, H-6a), 3.79 (t, *J* = 9.4 Hz, 1H, H-4), 3.72 (dd, *J* = 9.7 Hz, *J* = 15.0 Hz, 1H, H-6b), 3.65–3.38 (m, 6H, 3 × NC*H*_2_ DBU), 2.79–2.65 (m, 2H, C*H*_2_ DBU), 1.84–1.47 (m, 8H, 4 × C*H*_2_ DBU) ppm; ^13^C NMR (100 MHz, CDCl_3_) *δ* = 167.1 (1C, C_q_ DBU), 165.4 (1C, C_q_ Bz), 137.0, 134.6, 133.1, 132.9, 132.1, 129.1 (6C, 6 × C_q_ arom.), 133.6–126.1 (22C, arom.), 84.7 (1C, C-1), 78.2 (1C, C-3), 75.4 (1C, NAP*C*H_2_), 75.1 (1C, C-4), 71.7 (1C, C-5), 71.6 (1C, Bn*C*H_2_), 69.6 (1C, C-2), 55.6 (1C, N*C*H_2_ DBU), 55.3 (1C, C-6), 49.3, 48.2 (2C, 2 × N*C*H_2_ DBU), 29.0, 28.2, 25.6, 22.4, 19.7 (5C, 5 × *C*H_2_ DBU) ppm; UHR ESI-QTOF: *m/z* calcd for C_46_H_49_N_2_O_5_S [M]^+^ 741.3357; found: 741.3357.

*Phenyl 2,3-di-O-benzyl-4-O-(2′-naphthyl)methyl-1-thio-β-l-gulopyranoside* (**51**). Compound **40** (574 mg, 1.000 mmol) was converted to **51** according to general **Method C**. The crude product was purified by silica gel chromatography (65:35 *n*-hexane/EtOAc) to give **34** (70 mg, 12%) as a colorless syrup and **51** (385 mg, 65%) as a colorless syrup.

Data of **51**: [α]_D_^25^ + 3.1 (*c* 0.35, CHCl_3_); *R*_f_ 0.40 (65:35 *n*-hexane/EtOAc); ^1^H NMR (400 MHz, CDCl_3_) *δ* = 7.85–7.17 (m, 22H, arom.), 5.24 (d, *J* = 9.6 Hz, 1H, H-1), 4.68–4.33 (m, 6H, 2 × BnC*H*_2_, NAP*C*H_2_), 3.98–3.95 (m, 1H, H-5), 3.89–3.86 (m, 1H, H-6a), 3.80–3.76 (m, 2H, H-2, H-3), 3.53 (td, *J* = 4.2 Hz, *J* = 11.5 Hz, 1H, H-6b), 3.45 (dd, *J* = 1.2 Hz, *J* = 3.5 Hz, 1H, H-4), 1.78 (d, *J* = 8.5 Hz, 1H, H-6-OH) ppm; ^13^C NMR (100 MHz, CDCl_3_) *δ* = 138.0, 137.8, 135.1, 134.1, 133.0 (6C, 6 × C_q_ arom.), 131.5–125.8 (22C, arom.), 84.1 (1C, C-1), 76.0 (1C, C-5), 75.2 (1C, C-4), 74.8 (1C, C-2), 73.2, 72.8, 72,4 (3C, 2 × Bn*C*H_2_, NAP*C*H_2_), 72.6 (1C, C-3), 62.3 (1C, C-6) ppm; ESI-TOF-MS: *m/z* calcd for C_37_H_36_NaO_5_S [M + Na]^+^ 615.2176; found: 615.2183.

*Phenyl 2,3-di-O-benzoyl-4-O-(2′-naphthyl)methyl-1-thio-β-l-gulopyranoside* (**52**). Compound **45** (1.15 g, 1.909 mmol) was converted to **52** according to general **Method C**. The crude product was purified by silica gel chromatography (65:35 *n*-hexane/EtOAc) to give **35** (100 mg, 8%) as a colorless syrup and **52** (870 mg, 74%) as a colorless syrup.

Data of **52**: [α]_D_^25^ + 108.0 (*c* 0.10, CHCl_3_); *R*_f_ 0.32 (65:35 *n*-hexane/EtOAc); ^1^H NMR (500 MHz, CDCl_3_) *δ* = 8.01–7.26 (m, 22H, arom.), 6.02 (s, 1H, H-3), 5.57 (dd, *J* = 2.8 Hz, *J* = 10.3 Hz, 1H, H-2), 5.39 (d, *J* = 10.3 Hz, 1H, H-1), 5.07 (d, *J* = 11.9 Hz, 1H, NAPC*H*_2_a), 4.74 (d, *J* = 11.9 Hz, 1H, NAPC*H*_2_b), 4.10–4.08 (m, 1H, H-5), 3.99–3.96 (m, 1H, H-6a), 3.79 (d, *J* = 1.5 Hz, 1H, H-4), 3.60–3.55 (m, 1H, H-6b), 1.79 (d, *J* = 6.1 Hz, 1H, C-6-OH) ppm; ^13^C NMR (100 MHz, CDCl_3_) *δ* = 165.3, 165.1 (2C, 2 × C_q_ Bz), 134.5, 133.2, 132.3, 129.6, 129.3 (6C, 6 × C_q_ arom.), 133.7–126.0 (22C, arom.), 83.2 (1C, C-1), 77.1 (1C, C-5), 73.5 (1C, C-4), 72.4 (1C, NAP*C*H_2_), 67.8 (1C, C-2), 67.7 (1C, C-3), 62.3 (1C, C-6) ppm; ESI-TOF-MS: *m/z* calcd for C_37_H_32_NaO_7_S, [M + Na]^+^ 643.1761; found: 643.1743.

*Phenyl 2-O-benzoyl-3-O-benzyl-4-O-(2′-naphthyl)methyl-1-thio-β-l-gulopyranoside* (**53**). Compound **49** (325 mg, 0.553 mmol) was converted to **53** according to general **Method C**. The crude product was purified by silica gel chromatography (65:35 *n*-hexane/EtOAc) to give **36** (20 mg, 6%) as a colorless syrup and **53** (285 mg, 84%) as a colorless syrup.

Data of **53**: [α]_D_^25^ + 35.8 (*c* 0.19, CHCl_3_); *R*_f_ 0.41 (65:35 *n*-hexane/EtOAc); ^1^H NMR (500 MHz, CDCl_3_) *δ* = 8.09–7.10 (m, 22H, arom.), 5.42–5.36 (m, 2H, H-1, H-2), 4.80–4.43 (m, 4H, NAPC*H*_2_, BnC*H*_2_), 4.27–4.24 (m, 1H, H-3), 4.12–4.09 (m, 1H, H-5), 3.94 (dd, *J* = 7.6 Hz, *J* = 11.2 Hz, 1H, H-6a), 3.59 (d, *J* = 9.1 Hz, 1H, H-6b), 3.55 (s, 1H, H-4), 1.96 (s, 1H, C-6-OH) ppm; ^13^C NMR (125 MHz, CDCl_3_) *δ* = 165.5 (1C, C_q_ Bz), 137.5, 134.8, 133.2, 133.1, 129.9 (6C, 6 × C_q_ arom.), 133.4–126.0 (22C, arom.), 83.0 (1C, C-1), 76.5 (1C, C-5), 74.4 (1C, C-4), 73.7 (1C, NAP*C*H_2_), 72.9 (1C, C-3), 72.3 (1C, Bn*C*H_2_), 70.1 (1C, C-2), 62.6 (1C, C-6) ppm; ESI-TOF-MS: *m/z* calcd for C_37_H_34_NaO_6_S, [M + Na]^+^ 629.1968; found: 629.1961.

*Phenyl 2,3-di-O-benzyl-4,6-O-(2′-naphthyl)methylidene-1-thio-β-l-gulopyranoside* (**54**). Compound **51** (350 mg, 0.591 mmol) was converted to **54** according to general **Method G**. The crude product was purified by silica gel chromatography (7:3 *n*-hexane/acetone) to give **54** (267 mg, 77%) as a colorless syrup. [α]_D_^25^ + 7.5 (*c* 0.16, CHCl_3_); *R*_f_ 0.36 (8:2 *n*-hexane/EtOAc); ^1^H NMR (400 MHz, CDCl_3_) *δ* = 7.89–7.21 (m, 22H, arom.), 5.63 (s, 1H, H_ac_), 5.30 (d, *J* = 9.8 Hz, 1H, H-1), 4.83–4.60 (m, 5H, H-6a, 2 × BnC*H*_2_), 4.11 (d, *J* = 3.7 Hz, 1H, H-5), 4.04 (dd, *J* = 1.4 Hz, *J* = 12.4 Hz, 1H, H-6b), 3.95–3.93 (m, 1H, H-4), 3.88 (s, 1H, H-3), 3.85 (dd, *J* = 2.6 Hz, *J* = 10.0 Hz, 1H, H-2) ppm; ^13^C NMR (100 MHz, CDCl_3_) *δ* = 138.3, 138.1, 135.4, 133.9, 133.2, 133.0 (6C, 6 × C_q_ arom.), 132.9–124.2 (22C, arom.), 101.3 (1C, C_ac_), 83.4 (1C, C-1), 77.3 (1C, C-5), 74.6 (1C, C-4), 74.2 (1C, C-2), 73.8, 72.7 (2C, 2 × Bn*C*H_2_), 69.8 (1C, C-6), 67.7 (1C, C-3) ppm; ESI-TOF-MS: *m/z* calcd for C_37_H_34_NaO_5_S [M + Na]^+^ 613.2019; found: 613.2019.

*Phenyl 2,3-di-O-benzoyl-4,6-O-(2′-naphthyl)methylidene-1-thio-β-l-gulopyranoside* (**55**). Compound **52** (870 mg, 1.402 mmol) was converted to **55** according to general **Method G**. The crude product was purified by silica gel chromatography (65:35 *n*-hexane/EtOAc) to give **55** (719 mg, 83%) as a colorless syrup. [α]_D_^25^ + 35.0 (*c* 0.14, CHCl_3_); *R*_f_ 0.48 (65:35 *n*-hexane/EtOAc); ^1^H NMR (400 MHz, CDCl_3_) *δ* = 8.04–7.23 (m, 22H, arom.), 5.82 (s, 1H, H-3), 5.72 (s, 1H, H_ac_), 5.48 (dd, *J* = 2.7 Hz, *J* = 10.2 Hz, 1H, H-2), 5.40 (d, *J* = 10.2 Hz, 1H, H-1), 4.50 (d, *J* = 12.4 Hz, 1H, H-6a), 4.32 (d, *J* = 2.9 Hz, 1H, H-4), 4.15 (d, *J* = 12.3 Hz, 1H, H-6b), 4.01 (s, 1H, H-5) ppm; ^13^C NMR (125 MHz, CDCl_3_) *δ* = 165.0, 164.7 (2C, 2 × C_q_ Bz), 134.9, 134.0, 133.0, 130.9, 129.7, 129.5 (6C, 6 × C_q_ arom.), 134.2–124.2 (22C, arom.), 101.6 (1C, C_ac_), 82.2 (1C, C-1), 74.2 (1C, C-4), 69.6 (1C, C-3), 69.5 (1C, C-6), 68.5 (1C, C-5), 66.7 (1C, C-2) ppm; ESI-TOF-MS: *m/z* calcd for C_37_H_30_NaO_7_S [M + Na]^+^ 641.1604; found: 641.1606.

*Phenyl 2-O-benzoyl-3-O-benzyl-4,6-O-(2′-naphthyl)methylidene-1-thio-β-l-gulopyranoside* (**56**). Compound **53** (270 mg, 0.445 mmol) was converted to **56** according to general **Method G**. The crude product was purified by silica gel chromatography (85:15 CH_2_Cl_2_/*n*-hexane) to give **56** (179 mg, 67%) as a colorless syrup. [α]_D_^25^ + 40.9 (*c* 0.21, CHCl_3_); *R*_f_ 0.63 (85:15 CH_2_Cl_2_/*n*-hexane); ^1^H NMR (400 MHz, CDCl_3_) *δ* = 8.07–7.15 (m, 22H, arom.), 5.61 (s, 1H, H_ac_), 5.44 (dd, *J* = 2.4 Hz, *J* = 10.3 Hz, 1H, H-1), 5.33 (dt, *J* = 3.2 Hz, *J* = 10.3 Hz, 1H, H-2), 4.58 (s, 2H, BnC*H*_2_), 4.37 (d, *J* = 12.3 Hz, 1H, H-6a), 4.18 (s, 1H, H-3), 4.11 (d, *J* = 3.5 Hz, 1H, H-4), 4.01–3.98 (m, 1H, H-6b), 3.89 (s, 1H, H-5) ppm; ^13^C NMR (100 MHz, CDCl_3_) *δ* = 164.9 (1C, C_q_ Bz), 137.6, 135.2, 133.8, 132.8, 131.6 (6C, 6 × C_q_ arom.), 133.5–124.1 (22C, arom.) 101.2 (1C, C_ac_), 81.9 (1C, C-1), 75.0 (1C, C-3), 74.6 (1C, C-4), 73.8 (1C, Bn*C*H_2_), 69.5 (1C, C-6), 68.9 (1C, C-2), 67.9 (1C, C-5) ppm; ESI-TOF-MS: *m/z* calcd for C_37_H_32_NaO_6_S, [M + Na]^+^ 627.1812; found: 627.1822.

*Phenyl 2,3-di-O-benzyl-6-O-(2′-naphthyl)methyl-1-thio-β-l-gulopyranoside* (**57**). Compound **54** (250 mg, 0.423 mmol) was converted to **57** according to general **Method H**. The crude product was purified by silica gel chromatography (65:35 *n*-hexane/EtOAc) to give **57** (200 mg, 80%) as a colorless syrup. [α]_D_^25^ + 24.4 (*c* 0.18, CHCl_3_); *R*_f_ 0.46 (65:35 *n*-hexane/EtOAc); ^1^H NMR (400 MHz, CDCl_3_) *δ* = 7.79–7.16 (m, 22H, arom.), 5.26 (d, *J* = 10.0 Hz, 1H, H-1), 4.74–4.48 (m, 6H, 2 × BnC*H*_2_, NAPC*H*_2_), 4.06 (t, *J* = 3.8 Hz, 1H, H-5), 3.96 (s, 1H, H-4), 3.90 (t, *J* = 3.6 Hz, 1H, H-3), 3.83 (dd, *J* = 2.9 Hz, *J* = 9.9 Hz, 1H, H-2), 3.81 (dd, *J* = 3.4 Hz, *J* = 10.3 Hz, 1H, H-6a), 3.75 (dd, *J* = 4.4 Hz, *J* = 10.6 Hz, 1H, H-6b), 3.43 (s, 1H, C-4-OH) ppm; ^13^C NMR (100 MHz, CDCl_3_) *δ* = 138.2, 137.9, 134.9, 134.0, 133.1, 132.9 (6C, 6 × C_q_ arom.), 131.5–125.5 (22C, arom.), 84.7 (1C, C-1), 75.4 (1C, C-2), 74.7 (1C, C-3), 73.9 (1C, C-5), 73.7, 73.3, 72.5 (3C, 2 × Bn*C*H_2_, NAP*C*H_2_), 70.9 (1C, C-6), 70.0 (1C, C-4) ppm; ESI-TOF-MS: *m/z* calcd for C_37_H_36_NaO_5_S [M + Na]^+^ 615.2176; found: 615.2181.

*Phenyl 2,3-di-O-benzoyl-6-O-(2′-naphthyl)methyl-1-thio-β-l-gulopyranoside* (**58**). Compound **55** (90 mg, 0.145 mmol) was converted to **58** according to general **Method H**. The crude product was purified by silica gel chromatography (65:35 *n*-hexane/EtOAc) to give **58** (78 mg, 86%) as a colorless syrup. [α]_D_^25^ + 43.5 (*c* 0.17, CHCl_3_); *R*_f_ 0.34 (7:3 *n*-hexane/EtOAc); ^1^H NMR (400 MHz, CDCl_3_) *δ* = 7.99–7.21 (m, 22H, arom.), 5.77 (t, *J* = 3.5 Hz, 1H, H-3), 5.60 (dd, *J* = 3.3 Hz, *J* = 10.4 Hz, 1H, H-2), 5.36 (d, *J* = 10.4 Hz, 1H, H-1), 4.76 (d, *J* = 3.3 Hz, 2H, NAPC*H*_2_), 4.23 (t, *J* = 4.4 Hz, 1H, H-5), 4.14 (d, *J* = 3.5 Hz, 1H, H-4), 3.91 (d, *J* = 4.7 Hz, 2H, H-6a,b), 3.47 (s, 1H, H-4-OH) ppm; ^13^C NMR (100 MHz, CDCl_3_) *δ* = 165.1, 164.9 (2C, 2 × C_q_ Bz), 135.0, 133.3, 133.1, 132.5, 129.5, 129.4 (6C, 6 × C_q_ arom.), 133.6–125.8 (22C, arom.), 83.8 (1C, C-1), 75.5 (1C, C-5), 74.1 (1C, NAP*C*H_2_), 70.7 (1C, C-3), 70.3 (1C, C-6), 69.1 (1C, C-4), 67.1 (1C, C-2) ppm; ESI-TOF-MS: *m/z* calcd for C_37_H_32_NaO_7_S [M + Na]^+^ 643.1761; found: 643.1762.

*Phenyl 2-O-benzoyl-3-O-benzyl-6-O-(2′-naphthyl)methyl-1-thio-β-l-gulopyranoside* (**59**). Compound **56** (428 mg, 0.708 mmol) was converted to **59** according to general **Method H**. The crude product was purified by silica gel chromatography (65:35 *n*-hexane/EtOAc) to give **59** (373 mg, 87%) as a colorless syrup. [α]_D_^25^ + 20.4 (*c* 0.23, CHCl_3_); *R*_f_ 0.44 (65:35 *n*-hexane/EtOAc); ^1^H NMR (500 MHz, CDCl_3_) *δ* = 8.08–7.19 (m, 22H, arom.), 5.39 (dd, *J* = 2.7 Hz, *J* = 10.4 Hz, 1H, H-2), 5.36 (d, *J* = 10.4 Hz, 1H, H-1), 4.82–4.56 (m, 4H, NAPC*H*_2_, BnC*H*_2_), 4.20 (t, *J* = 4.2 Hz, 1H, H-5), 4.12–4.11 (m, 1H, H-3), 4.03 (t, *J* = 4.1 Hz, 1H, H-4), 3.90 (dd, *J* = 4.1 Hz, *J* = 10.6 Hz, 1H, H-6a), 3.85 (dd, *J* = 4.4 Hz, *J* = 10.6 Hz, 1H, H-6b), 3.23 (d, *J* = 4.3 Hz, 1H, C-4-O*H*) ppm; ^13^C NMR (100 MHz, CDCl_3_) *δ* = 165.1 (1C, C_q_ Bz), 137.8, 135.1, 133.1, 132.9, 129.7 (6C, 6 × C_q_ arom.), 133.2–125.6 (22C, arom.) 83.6 (1C, C-1), 76.5 (1C, C-3), 74.6 (1C, C-5), 73.8, 73.6 (2C, NAP*C*H_2_, Bn*C*H_2_), 70.6 (1C, C-6), 69.5 (1C, C-4), 69.1 (1C, C-2) ppm; ESI-TOF-MS: *m/z* calcd for C_37_H_34_NaO_6_S, [M + Na]^+^ 629.1968; found: 629.1954.

*Phenyl 2,3-di-O-benzyl-6-O-(2′-naphthyl)methyl-1-thio-β-l-ribo-hexopyranos-4-uloside* (**60**). Compound **57** (160 mg, 0.270 mmol) was converted to **60** according to general **Method I**. The crude product was purified by silica gel chromatography (7:3 *n*-hexane/EtOAc) to give **60** (107 mg, 67%) as a colorless syrup. [α]_D_^25^ + 1.2 (*c* 0.17, CHCl_3_); *R*_f_ 0.55 (7:3 *n*-hexane/EtOAc); ^1^H NMR (400 MHz, CDCl_3_) *δ* = 7.81–7.17 (m, 22H, arom.), 5.16 (d, *J* = 2.6 Hz, 1H, H-1), 4.73–4.54 (m, 6H, 2 × BnC*H*_2_, NAPC*H*_2_), 4.46 (d, *J* = 3.5 Hz, 1H, H-3), 4.32–4.29 (m, 1H, H-5), 4.19 (t, *J* = 3.0 Hz, 1H, H-2), 3.88 (qd, *J* = 3.4 Hz, *J* = 10.8 Hz, 2H, H-6a,b) ppm; ^13^C NMR (100 MHz, CDCl_3_) *δ* = 206.0 (1C, C-4), 137.4, 137.3, 135.5, 133.3, 133.1, 132.2 (6C, 6 × C_q_ arom.), 133.0–125.7 (22C, arom.), 86.4 (1C, C-1), 82.5 (1C, C-5), 82.1 (1C, C-3), 80.1 (1C, C-2), 73.7, 73.3, 72.8 (3C, 2 × Bn*C*H_2_, NAP*C*H_2_), 69.6 (1C, C-6) ppm; ESI-TOF-MS: *m/z* calcd for C_37_H_34_NaO_5_S [M + Na]^+^ 613.2019; found: 613.2011.

*Phenyl 2,3-di-O-benzoyl-6-O-(2′-naphthyl)methyl-1-thio-β-l-ribo-hexopyranos-4-uloside* (**61**). Compound **58** (69 mg, 0.112 mmol) was converted to **61** according to general **Method I**. The crude product was purified by silica gel chromatography (7:3 *n*-hexane/EtOAc) to give **61** (50 mg, 73%) as a colorless syrup. [α]_D_^25^ − 17.5 (*c* 0.12, CHCl_3_); *R*_f_ 0.34 (7:3 *n*-hexane/EtOAc); ^1^H NMR (400 MHz, CDCl_3_) *δ* = 7.98–7.24 (m, 22H, arom.), 6.12 (dd, *J* = 2.8 Hz, *J* = 4.2 Hz, 1H, H-2), 6.07 (d, *J* = 4.2 Hz, 1H, H-3), 5.31 (d, *J* = 2.7 Hz, 1H, H-1), 4.77 (s, 2H, NAPC*H*_2_), 4.51 (t, *J* = 3.8 Hz, 1H, H-5), 4.00 (d, *J* = 3.9 Hz, 2H, H-6a,b) ppm; ^13^C NMR (100 MHz, CDCl_3_) *δ* = 201.0 (1C, C-4), 165.0, 164.7 (2C, 2 × C_q_ Bz), 135.3, 133.4, 133.2, 131.3, 128.9 (6C, 6 × C_q_ arom.), 133.9–125.7 (22C, arom.), 85.9 (1C, C-1), 83.4 (1C, C-5), 74.7, 74.4 (2C, C-2, C-3), 73.9 (1C, NAP*C*H_2_), 69.5 (1C, C-6) ppm; ESI-TOF-MS: *m/z* calcd for C_37_H_30_NaO_7_S [M + Na]^+^ 641.1604; found: 641.1607.

*Phenyl 2-O-benzoyl-3-O-benzyl-6-O-(2′-naphthyl)methyl-1-thio-β-l-ribo-hexopyranos-4-uloside* (**62**). Compound **59** (342 mg, 0.563 mmol) was converted to **62** according to general **Method I**. The crude product was purified by silica gel chromatography (7:3 *n*-hexane/EtOAc) to give **62** (196 mg, 58%) as a colorless syrup. [α]_D_^25^ − 16.2 (*c* 0.08, CHCl_3_); *R*_f_ 0.42 (7:3 *n*-hexane/EtOAc); ^1^H NMR (400 MHz, CDCl_3_) *δ* = 7.97–7.16 (m, 22H, arom.), 5.85 (t, *J* = 4.0 Hz, 1H, H-2), 5.22 (d, *J* = 3.9 Hz, 1H, H-1), 4.67 (d, *J* = 2.3 Hz, 2H, NAPC*H*_2_), 4.55 (d, *J* = 12.2 Hz, 1H, BnC*H*_2_a), 4.46–4.40 (m, 3H, H-3, H-5, BnC*H*_2_b), 3.92 (d, *J* = 3.6 Hz, 2H, H-6) ppm; ^13^C NMR (100 MHz, CDCl_3_) *δ* = 203.9 (1C, C-4), 165.0 (1C, C_q_ Bz), 136.7, 135.2, 133.2, 133.0, 131.3, 129.1 (6C, 6 × C_q_ arom.), 133.6–125.6 (22C, arom.), 85.1 (1C, C-1), 82.4 (1C, C-5), 79.3 (1C, C-3), 73.8 (1C, C-2), 73.7, 72.2 (2C, NAP*C*H_2_, Bn*C*H_2_), 69.4 (1C, C-6) ppm; ESI-TOF-MS: *m/z* calcd for C_37_H_32_NaO_6_S, [M + Na]^+^ 627.1812; found: 627.1809.

*Phenyl 2,3-di-O-benzyl-6-O-(2′-naphthyl)methyl-1-thio-β-l-allopyranoside* (**63**). Compound **60** (100 mg, 0.169 mmol) was converted to **63** according to general **Method J**. The crude product was purified by silica gel chromatography (65:35 *n*-hexane/EtOAc) to give **63** (98 mg, 98%) as a colorless syrup. [α]_D_^25^ + 10.7 (*c* 0.29, CHCl_3_); *R*_f_ 0.33 (65:35 *n*-hexane/EtOAc); ^1^H NMR (500 MHz, CDCl_3_) *δ* = 7.78–7.19 (m, 22H, arom.), 5.20 (d, *J* = 7.1 Hz, 1H, H-1), 5.05 (d, *J* = 9.7 Hz, 1H, BnC*H*_2_a), 4.72–4.60 (m, 4H, NAPC*H*_2_, BnC*H*_2_), 4.56 (d, *J* = 10.8 Hz, 1H, BnC*H*_2_b), 4.05 (s, 1H, H-3), 3.83–3.81 (m, 2H, H-4, H-6a), 3.69–3.67 (m, 1H, H-6b), 3.54–3.51 (m, 1H, H-5), 3.44 (d, *J* = 8.3 Hz, 1H, H-2), 2.40 (s, 1H, C-4-OH) ppm; ^13^C NMR (125 MHz, CDCl_3_) *δ* = 138.4, 137.6, 135.9, 134.2, 133.3, 133.0 (6C, 6 × C_q_ arom.), 131.6–125.8 (22C, arom.), 84.0 (1C, C-1), 78.4 (1C, C-2), 77.5 (1C, C-3), 76.7 (1C, C-5), 75.3, 73.6, 73.1 (3C, 2 × Bn*C*H_2_, NAP*C*H_2_), 70.2 (1C, C-6), 68.1 (1C, C-4) ppm; ESI-TOF-MS: *m/z* calcd for C_37_H_36_NaO_5_S [M + Na]^+^ 615.2176; found: 615.2132.

*Phenyl 2,3-di-O-benzoyl-6-O-(2′-naphthyl)methyl-1-thio-β-l-gulopyranoside* (**58**) and *Phenyl 2,3-di-O-benzoyl-6-O-(2′-naphthyl)methyl-1-thio-β-l-allopyranoside* (**64**). Compound **61** (27 mg, 0.044 mmol) was converted to **64** according to general **Method J**. The crude product was purified by silica gel chromatography (7:3 n-hexane/EtOAc) to give mixture of **58** and **64** (22 mg, 81%, inseparable mixture of the l-gulo and the l-allo configured compounds, ratio of l-gulo (**58**): l-allo (**64**) = 5:1 based on the ^1^H NMR spectra (H-3 l-gulo: 5.75 ppm and H-3 l-allo: 6.10 ppm) as a colorless syrup.

Data of **64**: *R*_f_ 0.36 (7:3 *n*-hexane/EtOAc); ^1^H NMR (400 MHz, CDCl_3_) *δ* = 8.01–7.21 (m, 22H, arom.), 6.10 (t, *J* = 2.9 Hz, 1H, H-3), 5.27 (dd, *J* = 2.8 Hz, *J* = 10.2 Hz, 1H, H-2), 5.10 (d, *J* = 9.9 Hz, 1H, H-1), 4.78–4.67 (m, 2H, NAPC*H*_2_), 4.36 (ddd, *J* = 2.2 Hz, *J* = 5.4 Hz, *J* = 10.3 Hz, 1H, H-5), 3.82 (dd, *J* = 3.3 Hz, *J* = 6.5 Hz, 1H, H-4), 3.79 (dd, *J* = 2.2 Hz, *J* = 11.0 Hz, 1H, H-6a), 3.71 (dd, *J* = 5.5 Hz, *J* = 11.0 Hz, 1H, H-6b), 2.43 (d, *J* = 3.6 Hz, 1H, C-4-OH) ppm; ^13^C NMR (100 MHz, CDCl_3_) *δ* = 133.4–125.9 (22C, arom.), 85.5 (1C, C-1), 75.0 (1C, C-5), 74.1 (1C, NAP*C*H_2_), 70.3 (1C, C-3), 68.9 (1C, C-6), 68.1 (1C, C-4), 67.7 (1C, C-2) ppm; ESI-TOF-MS: *m/z* calcd for C_37_H_32_NaO_7_S [M + Na]^+^ 643.1761; found: 643.1828.

*Phenyl 2-O-benzoyl-3-O-benzyl-6-O-(2′-naphthyl)methyl-1-thio-β-l-allopyranoside* (**65**). Compound **62** (174 mg, 0.287 mmol) was converted to **65** according to general **Method J**. The crude product was purified by silica gel chromatography (7:3 *n*-hexane/EtOAc) to give **65** (155 mg, 89%) as a colorless syrup. [α]_D_^25^ + 25.9 (*c* 0.17, CHCl_3_); *R*_f_ 0.38 (7:3 *n*-hexane/EtOAc); ^1^H NMR (500 MHz, CDCl_3_) *δ* = 8.09–7.18 (m, 22H, arom.), 5.33 (d, *J* = 10.2 Hz, 1H, H-1), 4.99 (d, *J* = 10.0 Hz, 1H, H-2), 4.83 (d, *J* = 11.4 Hz, 1H, BnC*H*_2_a), 4.75 (s, 2H, NAPC*H*_2_), 4.56 (d, *J* = 11.4 Hz, 1H, BnC*H*_2_b), 4.30 (s, 1H, H-3), 3.94–3.93 (m, 1H, H-5), 3.88 (d, *J* = 10.5 Hz, 1H, H-6a), 3.79–3.72 (m, 2H, H-4, H-6b), 2.44 (d, *J* = 9.3 Hz, 1H, C-4-O*H*) ppm; ^13^C NMR (125 MHz, CDCl_3_) *δ* = 165.3 (1C, C_q_ Bz), 137.9, 135.7, 133.4, 133.1, 132.7, 129.6 (6C, 6 × C_q_ arom.), 133.6–125.8 (22C, arom.), 82.5 (1C, C-1), 77.9 (1C, C-3), 76.9 (1C, C-5), 75.8, 73.8 (2C, NAP*C*H_2_, Bn*C*H_2_), 71.7 (1C, C-2), 70.4 (1C, C-6), 68.5 (1C, C-4) ppm; ESI-TOF-MS: *m/z* calcd for C_37_H_34_NaO_6_S, [M + Na]^+^ 629.1968; found: 629.1964.

#### 3.2.13. Synthesis of l-Galactose and l-Glucose Derivatives from d-Altrose

*Phenyl 2-O-benzoyl-4,6-O-(2′-naphthyl)methylidene-1-thio-α-d-mannopyranoside* (**66**). To a solution of compound **29** [91] (2.5 g, 6.096 mmol) in dry CH_3_CN (85 mL), we added triethyl-orthobenzoate (2.22 mL, 9.753 mmol, 1.6 equiv.) and stirred for 15 min. After 15 min at that temperature, CSA (425 mg, 1.828 mmol, 0.30 equiv.) was added and the mixture was stirred for 2 h at room temperature. After that, the solvents were evaporated. The residue was dissolved in AcOH (80%, 40 mL) at 0 °C and stirred for 15 min. The reaction mixture was neutralized with NaHCO_3_, extracted with CH_2_Cl_2_ (3 × 100 mL), and the organic phases was washed with H_2_O (3 × 50 mL) until neutral pH. The organic layer was dried over MgSO_4_ and concentrated. The crude product was purified by silica gel chromatography (7:3 *n*-hexane/acetone) to give **66** (1.81 g, 57%) as a colorless syrup. [α]_D_^25^ −6.2 (*c* 0.13, CHCl_3_); *R*_f_ 0.42 (7:3 *n*-hexane/acetone); ^1^H NMR (500 MHz, CDCl_3_) *δ* = 8.11–7.24 (m, 17H, arom.), 5.83 (s, 1H, H_ac_), 5.75 (dd, *J* = 1.3 Hz, *J* = 3.5 Hz, 1H, H-2), 5.64 (d, *J* = 1.0 Hz, 1H, H-1), 4.49 (td, *J* = 4.9 Hz, *J* = 9.8 Hz, 1H, H-5), 4.39 (dt, *J* = 3.7 Hz, *J* = 9.8 Hz, 1H, H-3), 4.33 (dd, *J* = 4.9 Hz, *J* = 10.3 Hz, 1H, H-6a), 4.18 (t, *J* = 9.7 Hz, 1H, H-4), 3.95 (t, *J* = 10.3 Hz, 1H, H-6b), 2.56 (dd, *J* = 4.1 Hz, *J* = 9.4 Hz, 1H, C-3-O*H*) ppm; ^13^C NMR (125 MHz, CDCl_3_) *δ* = 166.1 (1C, C_q_ Bz), 134.5, 133.9, 133.2, 133.1, 129.6 (5C, 5 × C_q_ arom.), 133.7–123.8 (17C, arom.), 102.6 (1C, C_ac_), 87.2 (1C, C-1), 79.8 (1C, C-4), 74.3 (1C, C-2), 68.7 (1C, C-6), 68.3 (1C, C-3), 64.9 (1C, C-5) ppm; ESI-TOF-MS: *m/z* calcd for: C_30_H_26_NaO_6_S [M + Na]^+^ 537.1342; found: 537.1348.

*Phenyl 2-O-benzoyl-4,6-O-(2′-naphthyl)methylidene-1-thio-α-d-altropyranoside* (**67**). Compound **66** (5.68 g, 11.05 mmol) was converted to **3-ulose** according to general **Method I**. The crude product was purified by silica gel chromatography (6:4 *n*-hexane/EtOAc) to give the **3-ulose** derivative (4.37 g, 77%) as a colorless syrup. [α]_D_^25^ + 112.0 (*c* 0.15, CHCl_3_); *R*_f_ 0.39 (7:3 *n*-hexane/EtOAc); ^1^H NMR (400 MHz, CDCl_3_) *δ* = 8.15–7.20 (m, 17H, arom.), 5.92 (s, 1H, H-1), 5.81 (s, 1H, H_ac_), 5.54 (s, 1H, H-2), 4.98 (d, *J* = 9.7 Hz, 1H, H-4), 4.77–4.75 (m, 1H, H-5), 4.41–4.30 (m, 1H, H-6a), 4.05 (t, *J* = 10.0 Hz, 1H, H-6b) ppm; ^13^C NMR (100 MHz, CDCl_3_) *δ* = 193.4 (1C, C-3), 164.8 (1C, C_q_ Bz), 133.8, 133.7, 132.9, 131.4 (5C, 5 × C_q_ arom.), 134.1–123.8 (17C, arom.), 102.2 (1C, C_ac_), 88.6 (1C, C-1), 81.9 (1C, C-4), 78.6 (1C, C-2), 69.1 (1C, C-6), 68.1 (1C, C-5) ppm; ESI-TOF-MS: *m/z* calcd for: C_30_H_24_NaO_6_S [M + Na]^+^ 535.1186; found: 535.1210.

To the solution of the **3-ulose** compound (4.44 g, 8.669 mmol) in dry MeOH (53.6 mL), NaBH_4_ (492 mg, 13.00 mmol, 1.5 equiv.) was added and the reaction mixture was stirred for 1 h at room temperature. After 1 h the mixture was neutralized with 60% AcOH (6 mL) and concentrated, then the residue was coevaporated with MeOH (3 × 100 mL). The crude product was purified by silica gel chromatography (CH_2_Cl_2_) to give **67** (2.61 g, 58%) as a white foam and **66** (710 mg, 16%) as a colorless syrup.

Data of **67**: [α]_D_^25^ +167.0 (*c* 0.10, CHCl_3_); *R*_f_ 0.40 (CH_2_Cl_2_); ^1^H NMR (500 MHz, CDCl_3_) *δ* = 8.08–7.25 (m, 17H, arom.), 5.90 (s, 1H, H_ac_), 5.62 (d, *J* = 3.2 Hz, 1H, H-2), 5.55 (s, 1H, H-1), 4.93 (td, *J* = 5.1 Hz, *J* = 10.1 Hz, 1H, H-5), 4.44 (dd, *J* = 5.1 Hz, *J* = 10.3 Hz, 1H, H-6a), 4.38 (s, 1H, H-3), 4.17 (dd, *J* = 2.9 Hz, *J* = 9.8 Hz, 1H, H-4), 3.97 (t, *J* = 10.4 Hz, 1H, H-6b), 2.58 (d, *J* = 1.7 Hz, 1H, C-3-O*H*) ppm; ^13^C NMR (100 MHz, CDCl_3_) *δ* = 165.0 (1C, C_q_ Bz), 136.1, 134.5, 133.9, 133.0, 129.3 (5C, 5 × C_q_ arom.), 133.7–123.8 (17C, arom.), 102.5 (1C, C_ac_), 86.3 (1C, C-1), 77.4 (1C, C-4), 74.1 (1C, C-2), 69.1 (1C, C-6), 66.9 (1C, C-3), 59.6 (1C, C-5) ppm; ESI-TOF-MS: *m/z* calcd for: C_30_H_26_NaO_6_S [M + Na]^+^ 537.1342; found: 537.1389.

*Phenyl 2,3-di-O-benzyl-4,6-O-(2′-naphthyl)methylidene-1-thio-α-d-altropyranoside* (**68**). To the solution of compound **67** (888 mg, 1.728 mmol) in MeOH (21 mL), NaOMe (55 mg) was added, and the reaction mixture was stirred for 24 h at room temperature. After 24 h the reaction mixture was neutralized by Amberlite IR-120 (H^+^) ion-exchange resin, then it was filtered, washed with MeOH, and concentrated. The crude product was reacted in the further reaction without purification (*R*_f_ 0.35 (6:4 *n*-hexane/EtOAc)). To a solution of the crude product (709 mg, 1.728 mmol), dry DMF (7.3 mL) at 0 °C NaH (60%, 173 mg, 4.320 mmol, 1.25 equiv./-OH) was added in portions. After 30 min stirring at this temperature, BnBr (513 μL, 4.320 mmol, 1.25 equiv./-OH) was added, and the reaction mixture was allowed to warm up to room temperature and stirred for 24 h. After completion of the reaction, MeOH (20 mL) was added. The reaction mixture was stirred for 15 min, then the solvents were evaporated. The residue was diluted with CH_2_Cl_2_ (500 mL), washed with H_2_O (3 × 100 mL), dried over MgSO_4_, filtered, and concentrated. The crude product was purified by silica gel chromatography (8:2 *n*-hexane/EtOAc) to give **68** (796 mg, 78% for two steps) as a yellow syrup. [α]_D_^25^ −145.0 (*c* 0.12, CHCl_3_); *R*_f_ 0.49 (8:2 *n*-hexane/EtOAc); ^1^H NMR (500 MHz, CDCl_3_) *δ* = 7.96–7.25 (m, 22H, arom.), 5.75 (s, 1H, H_ac_), 5.49 (s, 1H, H-1), 4.88–4.85 (m, 2H, H-5, BnC*H*_2_a), 4.74 (d, *J* = 12.5 Hz, 1H, BnC*H*_2_b), 4.61 (d, *J* = 11.9 Hz, 1H, BnC*H*_2_a), 4.52 (d, *J* = 11.9 Hz, 1H, BnC*H*_2_b), 4.37 (dd, *J* = 5.0 Hz, *J* = 10.0 Hz, 1H, H-6a), 4.18 (d, *J* = 9.5 Hz, 1H, H-4), 4.03–4.01 (m, 2H, H-2, H-3), 3.89 (t, *J* = 10.3 Hz, 1H, H-6b) ppm; ^13^C NMR (125 MHz, CDCl_3_) *δ* = 138.5, 137.4, 137.3, 135.3, 133.8, 133.1 (6C, 6 × C_q_ arom.), 130.8–124.0 (22C, arom.), 102.5 (1C, C_ac_), 86.4 (1C, C-1), 78.9 (1C, C-2), 77.7 (1C, C-4), 73.5 (1C, C-3), 73.2, 72.4 (2C, 2 × Bn*C*H_2_), 69.4 (1C, C-6), 60.3 (1C, C-5) ppm; ESI-TOF-MS: *m/z* calcd for: C_37_H_34_NaO_5_S [M + Na]^+^ 613.2019; found: 613.2014.

*Phenyl 2,3-di-O-benzoyl-4,6-O-(2′-naphthyl)methylidene-1-thio-α-d-altropyranoside* (**69**). To a stirred solution of compound **67** (892 mg, 1.735 mmol) in dry pyridine (4.7 mL), BzCl (252 µL, 2.169 mmol, 1.25 equiv.) was added at 0 °C and the reaction mixture was stirred for 24 h at room temperature. After 24 h the mixture was diluted with CH_2_Cl_2_ (100 mL), washed with H_2_O (25 mL), 1M aqueous solution of H_2_SO_4_ (2 × 25 mL), H_2_O (25 mL), saturated aqueous solution of NaHCO_3_ (2 × 25 mL), and H_2_O (25 mL) until neutral pH. The organic layer was separated, dried over MgSO_4_, filtered, and concentrated under reduced pressure. The crude product was purified by silica gel chromatography (7:3 → 8:2 CH_2_Cl_2_/*n*-hexane) to give **69** (887 mg, 83%) as a colorless syrup. [α]_D_^25^ + 174.2 (*c* 0.12, CHCl_3_); *R*_f_ 0.49 (7:3 CH_2_Cl_2_/*n*-hexane); ^1^H NMR (500 MHz, CDCl_3_) *δ* = 8.34–7.23 (m, 22H, arom.), 5.88 (s, 1H, H_ac_), 5.83 (s, 1H, H-3), 5.72 (s, 1H, H-2), 5.61 (s, 1H, H-1), 5.05 (td, *J* = 5.2 Hz, *J* = 9.9 Hz, 1H, H-5), 4.47 (dd, *J* = 5.0 Hz, *J* = 10.3 Hz, 1H, H-6a), 4.38 (d, *J* = 9.6 Hz, 1H, H-4), 3.99 (t, *J* = 10.4 Hz, 1H, H-6b) ppm; ^13^C NMR (125 MHz, CDCl_3_) *δ* = 165.4, 164.9 (2C, 2 × C_q_ Bz), 135.2, 134.5, 133.8, 133.0, 129.5, 129.2 (6C, 6 × C_q_ arom.), 133.9–123.7 (22C, arom.), 102.3 (1C, C_ac_), 86.5 (1C, C-1), 75.6 (1C, C-4), 72.6 (1C, C-2), 69.3 (1C, C-6), 67.3 (1C, C-3), 60.7 (1C, C-5) ppm; ESI-TOF-MS: *m/z* calcd for: C_37_H_30_NaO_7_S [M + Na]^+^ 641.1604; found: 641.1608.

*Phenyl 2,3-di-O-benzyl-4-O-(2′-naphthyl)methyl-1-thio-α-d-altropyranoside* (**70**). Compound **68** (750 mg, 1.270 mmol) was converted to **70** according to general **Method D**. The crude product was purified by silica gel chromatography (65:35 *n*-hexane/EtOAc) to give **70** (660 mg, 88%) as a colorless syrup. [α]_D_^25^ + 140.0 (*c* 0.17, CHCl_3_); *R*_f_ 0.48 (65:35 *n*-hexane/EtOAc); ^1^H NMR (500 MHz, CDCl_3_) *δ* = 7.83–7.12 (m, 22H, arom.), 5.44 (s, 1H, H-1), 4.73–4.33 (m, 7H, H-5, NAPC*H*_2_, 2 × BnC*H*_2_), 3.96–3.94 (m, 2H, H-2, H-4), 3.93–3.88 (m, 2H, H-6a,b), 3.83 (s, 1H, H-4), 1.89 (s, 1H, C-6-O*H*) ppm; ^13^C NMR (125 MHz, CDCl_3_) *δ* = 138.0, 137.5, 137.4, 135.6, 133.4, 133.1 (6C, 6 × C_q_ arom.), 131.1–126.0 (22C, arom.), 86.2 (1C, C-1), 77.4 (1C, C-2), 72.8 (1C, C-4), 72.6 (1C, C-3), 72.2, 72.2, 71.7 (3C, NAP*C*H_2_, 2 × Bn*C*H_2_), 68.6 (1C, C-5), 62.7 (1C, C-6) ppm; ESI-TOF-MS: *m/z* calcd for: C_37_H_36_NaO_5_S [M + Na]^+^ 615.2176; found: 615.2173.

*Phenyl 2,3-di-O-benzoyl-4-O-(2′-naphthyl)methyl-1-thio-α-d-altropyranoside* (**71**). Compound **69** (823 mg, 1.331 mmol) was converted to **71** according to general **Method D**. The crude product was purified by silica gel chromatography (65:35 *n*-hexane/EtOAc) to give **71** (728 mg, 88%) as a colorless syrup. [α]_D_^25^ +131.8 (*c* 0.11, CHCl_3_); *R*_f_ 0.49 (6:4 *n*-hexane/EtOAc); ^1^H NMR (500 MHz, CDCl_3_) *δ* = 8.35–7.25 (m, 22H, arom.), 5.94 (s, 1H, H-3), 5.59 (d, *J* = 3.1 Hz, 1H, H-2), 5.56 (s, 1H, H-1), 4.93 (d, *J* = 11.5 Hz, 1H, NAPC*H*_2_a), 4.77–4.72 (m, 2H, H-5, NAPC*H*_2_b), 4.14 (dd, *J* = 2.7 Hz, *J* = 9.9 Hz, 1H, H-4), 3.97 (d, *J* = 11.6 Hz, 1H, H-6a), 3.91–3.87 (m, 1H, H-6b), 1.86 (t, *J* = 6.2 Hz, 1H, C-6-O*H*) ppm; ^13^C NMR (125 MHz, CDCl_3_) *δ* = 165.6, 164.9 (2C, 2 × C_q_ Bz), 135.4, 134.7, 133.3, 133.2, 129.5, 129.1 (6C, 6 × C_q_ arom.), 133.7–126.2 (22C, arom.), 86.1 (1C, C-1), 72.5 (1C, C-2), 71.5 (1C, NAP*C*H_2_), 70.5 (1C, C-4), 68.6 (1C, C-5), 66.1 (1C, C-3), 62.5 (1C, C-6) ppm; ESI-TOF-MS: *m/z* calcd for: C_37_H_32_NaO_7_S [M + Na]^+^ 643.1761; found: 643.1821.

*Phenyl 2,3-di-O-benzyl-6-deoxy-6-iodo-4-O-(2′-naphthyl)methyl-1-thio-α-d-altropyranoside* (**72**). Compound **70** (650 mg, 1.095 mmol) was converted to **72** according to general **Method A**. The crude product was purified by silica gel chromatography (4:6 → 8:2 CH_2_Cl_2_/*n*-hexane) to give **72** (665 mg, 87%) as a white foam. [α]_D_^25^ + 93.3 (*c* 0.12, CHCl_3_); *R*_f_ 0.57 (7:3 CH_2_Cl_2_/*n*-hexane); ^1^H NMR (500 MHz, CDCl_3_) *δ* = 7.85–7.19 (m, 22H, arom.), 5.51 (s, 1H, H-1), 4.72–4.48 (m, 5H, NAPC*H*_2_, 2 × BnC*H*_2_a, BnC*H*_2_b), 4.40 (t, *J* = 6.8 Hz, 1H, H-5), 4.36 (d, *J* = 12.2 Hz, 1H, BnC*H*_2_b), 4.01 (d, *J* = 1.9 Hz, 1H, H-2), 3.81–3.79 (m, 2H, H-3, H-4), 3.64–3.62 (m, 1H, H-6a), 3.47 (dd, *J* = 6.5 Hz, *J* = 10.6 Hz, 1H, H-6b) ppm; ^13^C NMR (125 MHz, CDCl_3_) *δ* = 137.8, 137.7, 137.6, 135.3, 133.3, 133.1 (6C, 6 × C_q_ arom.), 131.0–126.1 (22C, arom.), 86.2 (1C, C-1), 77.2 (1C, C-2), 76.7 (1C, C-3), 72.5, 72.0, 71.6 (3C, NAP*C*H_2_, 2 × Bn*C*H_2_), 72.1 (1C, C-4), 67.5 (1C, C-5), 8.7 (1C, C-6) ppm; ESI-TOF-MS: *m/z* calcd for: C_37_H_35_INaO_4_S [M + Na]^+^ 725.1193; found: 725.1195.

*Phenyl 2,3-di-O-benzoyl-6-deoxy-6-iodo-4-O-(2′-naphthyl)methyl-1-thio-α-d-altropyranoside* (**73**). Compound **71** (676 mg, 1.089 mmol) was converted to **73** according to general **Method A**. The crude product was purified by silica gel chromatography (1:1 → 8:2 CH_2_Cl_2_/*n*-hexane) to give **73** (759 mg, 95%) as a colorless syrup. [α]_D_^25^ + 99.2 (*c* 0.13, CHCl_3_); *R*_f_ 0.46 (8:2 *n*-hexane/EtOAc); ^1^H NMR (500 MHz, CDCl_3_) *δ* = 8.34–7.26 (m, 22H, arom.), 5.95 (s, 1H, H-3), 5.64 (s, 1H, H-2), 5.59 (s, 1H, H-1), 4.96 (d, *J* = 11.1 Hz, 1H, NAPC*H*_2_a), 4.72 (d, *J* = 11.1 Hz, 1H, NAPC*H*_2_b), 4.46–4.43 (m, 1H, H-5), 4.02 (d, *J* = 9.2 Hz, 1H, H-4), 3.64 (d, *J* = 10.6 Hz, 1H, H-6a), 3.52 (dd, *J* = 5.7 Hz, *J* = 10.5 Hz, 1H, H-6b) ppm; ^13^C NMR (125 MHz, CDCl_3_) *δ* = 165.5, 164.8 (2C, 2 × C_q_ Bz), 135.6, 134.4, 133.3, 129.3, 129.1 (6C, 6 × C_q_ arom.), 133.7–126.3 (22C, arom.), 86.3 (1C, C-1), 74.5 (1C, C-4), 72.4 (1C, C-2), 71.7 (1C, NAP*C*H_2_), 67.3 (1C, C-5), 65.6 (1C, C-3), 8.3 (1C, C-6) ppm; ESI-TOF-MS: *m/z* calcd for: C_37_H_31_INaO_6_S [M + Na]^+^ 753.0778; found: 753.0792.

*Phenyl 2,3-di-O-benzyl-4-O-(2′-naphthyl)methyl-1-thio-α-d-arabino-hex-5-enopyranoside* (**74**).

**Reaction I.**: Compound **72** (218 mg, 0.310 mmol) was converted to **74** according to general **Method E**. The crude product was purified by silica gel chromatography (7:3 CH_2_Cl_2_/*n*-hexane) to give **74** (151 mg, 85%) as a colorless syrup.

**Reaction II.**: Compound **72** (201 mg, 0.286 mmol) was converted to **74** according to general **Method F**. The crude product was purified by silica gel chromatography (7:3 CH_2_Cl_2_/*n*-hexane) to give **72** (31 mg, 19%) as a colorless syrup and **74** (85 mg, 52%) as a colorless syrup.

Data of **74**: [α]_D_^25^ + 50.8 (*c* 0.12, CHCl_3_); *R*_f_ 0.47 (8:2 CH_2_Cl_2_/*n*-hexane); ^1^H NMR (500 MHz, CDCl_3_) *δ* = 7.85–7.20 (m, 22H, arom.), 4.96 (d, *J* = 7.5 Hz, 1H, H-1), 4.91 (s, 1H, H-6a), 4.91–4.59 (m, 6H, NAPC*H*_2_, 2 × BnC*H*_2_), 4.53 (s, 1H, H-6b), 4.13–4.10 (m, 2H, H-2, H-4), 3.67 (dd, *J* = 3.1 Hz, *J* = 8.0 Hz, 1H, H-3) ppm; ^13^C NMR (125 MHz, CDCl_3_) *δ* = 153.9 (1C, C-5), 138.2, 138.1, 135.4, 134.3, 133.4, 133.1 (6C, 6 × C_q_ arom.), 131.9–126.0 (22C, arom.), 100.2 (1C, C-6), 88.6 (1C, C-1), 80.0 (1C, C-3), 77.1 (1C, C-2), 74.9, 72.1, 69.9 (3C, NAP*C*H_2_, 2 × Bn*C*H_2_), 73.3 (1C, C-4) ppm; ESI-TOF-MS: *m/z* calcd for: C_37_H_34_NaO_4_S [M + Na]^+^ 597.2070; found: 597.2051.

*Phenyl 2,3-di-O-benzyl-4-O-(2′-naphthyl)methyl-1-thio-α-d-arabino-hex-5-enopyranoside* (**74**) and *8-N-[phenyl 2,3-di-O-benzyl-6-deoxy-6-yl-4-O-(2′-naphthyl)methyl-1-thio-α-d-altropyranoside]-1,8-diazabicyclo(5.4.0)undec-7-ene-iodide* (**75**). Compound **72** (215 mg, 0.306 mmol) was converted to **74** according to general **Method B**. The crude product was purified by silica gel chromatography (7:3 CH_2_Cl_2_/n-hexane → 95:5 CH_2_Cl_2_/MeOH) to give **74** (102 mg, 58%) as a colorless syrup and **75** (107 mg, 41%) as a light yellow syrup.

Data of **75**: [α]_D_^25^ + 120.8 (*c* 0.13, CHCl_3_); *R*_f_ 0.44 (7:3 CH_2_Cl_2_/MeOH); ^1^H NMR (500 MHz, CDCl_3_) *δ* = 7.86–7.17 (m, 22H, arom.), 5.51 (s, 1H, H-1), 4.69–4.41 (m, 7H, H-5, NAPC*H*_2_, 2 × BnC*H*_2_), 4.00 (dd, *J* = 1.6 Hz, *J* = 15.3 Hz, 1H, H-6a), 3.94 (d, *J* = 3.4 Hz, 1H, H-2), 3.86 (t, *J* = 3.0 Hz, 1H, H-3), 3.68 (dd, *J* = 2.8 Hz, *J* = 9.9 Hz, 1H, H-4), 3.64–3.39 (m, 7H, H-6b, 3 × NC*H*_2_ DBU), 2.84–2.68 (m, 2H, C*H*_2_ DBU), 1.84–1.80 (m, 2H, C*H*_2_ DBU), 1.61–1.47 (m, 6H, 3 × C*H*_2_ DBU) ppm; ^13^C NMR (125 MHz, CDCl_3_) *δ* = 167.2 (1C, C_q_ DBU), 137.2, 137.0, 136.1, 134.4, 133.0, 132.9 (6C, 6 × C_q_ arom.), 129.4–125.9 (22C, arom.), 84.5 (1C, C-1), 76.4 (1C, C-2), 73.4 (1C, C-4), 72.5, 72.3, 70.9 (3C, NAP*C*H_2_, 2 × Bn*C*H_2_), 71.0 (1C, C-3), 66.5 (1C, C-5), 55.5 (1C, C-6), 55.0, 49.4, 48.0 (3C, 3 × N*C*H_2_ DBU), 28.9, 28.1, 25.6, 22.5, 19.7 (5C, 5 × C*H*_2_ DBU) ppm; UHR ESI-QTOF: *m/z* calcd for C_46_H_51_N_2_O_4_S [M]^+^ 727.3564; found: 727.3572.

*Phenyl 4-O-(2′-naphthyl)methyl-1-thio-α-d-arabino-hex-5-enopyranoside* (**76**) and *Phenyl 2,6-anhydro-4-O-(2′-naphthyl)methyl-1-thio-α-d-altropyranoside* (**77**). Compound **73** (254 mg, 0.347 mmol) was converted to **76** according to general **Method E**. The crude product was purified by silica gel chromatography (97:3 → 9:1 CH_2_Cl_2_/acetone) to give **76** (60 mg, 43%) as a white foam and **77** (30 mg, 22%) as a white foam.

Data of **76:** [α]_D_^25^ +34.3 (*c* 0.14, CHCl_3_); *R*_f_ 0.17 (97:3 CH_2_Cl_2_/acetone); ^1^H NMR (500 MHz, CDCl_3_) *δ* = 7.81–7.21 (m, 12H, arom.), 4.94 (s, 1H, H-6a), 4.77 (d, *J* = 11.7 Hz, 1H, NAPC*H*_2_a), 4.64 (d, *J* = 9.3 Hz, 1H, H-1), 4.55 (s, 1H, H-6b), 4.43 (d, *J* = 11.7 Hz, 1H, NAPC*H*_2_b), 4.03 (d, *J* = 3.5 Hz, 1H, H-4), 3.87 (t, *J* = 9.1 Hz, 1H, H-2), 3.65 (d, *J* = 6.3 Hz, 1H, H-3), 3.06 (s, 2H, C-2-O*H*, C-3-O*H*) ppm; ^13^C NMR (125 MHz, CDCl_3_) *δ* = 154.0 (1C, C-5), 134.9, 133.3, 133.1, 132.0 (4C, 4 × C_q_ arom.), 132.7–125.9 (12C, arom.), 102.0 (1C, C-6), 88.9 (1C, C-1), 76.7 (1C, C-4), 73.6 (1C, C-3), 70.1 (1C, NAP*C*H_2_), 69.5 (1C, C-2) ppm; ESI-TOF-MS: *m/z* calcd for: C_23_H_22_NaO_4_S [M + Na]^+^ 417.1131; found: 417.1131.

Data of **77:** [α]_D_^25^ +227.3 (*c* 0.12, CHCl_3_); *R*_f_ 0.54 (97:3 CH_2_Cl_2_/acetone); ^1^H NMR (500 MHz, CDCl_3_) *δ* = 7.86–7.23 (m, 12H, arom.), 5.72 (t, *J* = 1.5 Hz, 1H, H-1), 4.93 (s, 2H, NAPC*H*_2_), 4.28–4.24 (m, 1H, H-5), 4.18 (dd, *J* = 2.7 Hz, *J* = 10.4 Hz, 1H, H-6a), 4.07 (s, 1H, H-3), 4.05 (dd, *J* = 1.9 Hz, *J* = 3.6 Hz, 1H, H-2), 3.88 (dd, *J* = 2.0 Hz, *J* = 8.8 Hz, 1H, H-4), 3.73 (s, 1H, C-3-O*H*), 3.71 (d, *J* = 4.2 Hz, 1H, H-6b) ppm; ^13^C NMR (125 MHz, CDCl_3_) *δ* = 136.5, 134.6, 133.3, 133.2 (4C, 4 × C_q_ arom.), 131.2–125.8 (12C, arom.), 85.8 (1C, C-1), 72.5 (1C, NAP*C*H_2_), 72.2 (1C, C-4), 70.2 (1C, C-2), 68.0 (1C, C-3), 66.3 (1C, C-6), 66.1 (1C, C-5) ppm; ESI-TOF-MS: *m/z* calcd for: C_23_H_22_NaO_4_S [M + Na]^+^ 417.1131; found: 417.1144.

*Phenyl 2,3-di-O-benzoyl-4-O-(2′-naphthyl)methyl-1-thio-α-d-arabino-hex-5-enopyranoside* (**78**). To a stirred solution of compound **76** (60 mg, 0.150 mmol) in dry pyridine (1.0 mL), BzCl (52 µL, 0.449 mmol, 1.5 equiv.) was added at 0 °C and the reaction mixture was stirred for 24 h at room temperature. After 24 h the mixture was diluted with CH_2_Cl_2_ (50 mL), washed with H_2_O (15 mL), 1M aqueous solution of H_2_SO_4_ (2 × 15 mL), H_2_O (15 mL), saturated aqueous solution of NaHCO_3_ (2 × 15 mL), and H_2_O (15 mL) until neutral pH. The organic layer was separated, dried over MgSO_4_, filtered, and concentrated under reduced pressure. The crude product was purified by silica gel chromatography (8:2 CH_2_Cl_2_/*n*-hexane) to give **78** (57 mg, 63%) as a colorless syrup. [α]_D_^25^ + 10.8 (*c* 0.12, CHCl_3_); *R*_f_ 0.45 (7:3 CH_2_Cl_2_/*n*-hexane); ^1^H NMR (500 MHz, CDCl_3_) *δ* = 8.08–7.23 (m, 22H, arom.), 5.96 (dd, *J* = 7.2 Hz, *J* = 7.8 Hz, 1H, H-2), 5.48 (dd, *J* = 3.2 Hz, *J* = 8.0 Hz, 1H, H-3), 5.25 (d, *J* = 7.0 Hz, 1H, H-1), 5.05 (s, 1H, H-6a), 4.87 (d, *J* = 12.6 Hz, 1H, NAPC*H*_2_a), 4.81 (s, 1H, H-6b), 4.63 (d, *J* = 12.6 Hz, 1H, NAPC*H*_2_b), 4.43 (d, *J* = 3.2 Hz, 1H, H-4) ppm; ^13^C NMR (125 MHz, CDCl_3_) *δ* = 165.8, 165.1 (2C, 2 × C_q_ Bz), 152.8 (1C, C-5), 135.0, 133.3, 133.1, 129.4 (6C, 6 × C_q_ arom.), 133.5–125.7 (22C, arom.), 101.3 (1C, C-6), 87.6 (1C, C-1), 73.1 (1C, C-4), 72.1 (1C, C-3), 70.3 (1C, NAP*C*H_2_), 69.3 (1C, C-2) ppm; ESI-TOF-MS: *m/z* calcd for: C_37_H_30_NaO_6_S [M + Na]^+^ 625.1655; found: 625.1654.

*Phenyl 2,3-di-O-benzoyl-4-O-(2′-naphthyl)methyl-1-thio-α-d-arabino-hex-5-enopyranoside* (**78**) and *8-N-[phenyl 2,3-di-O-benzoyl-6-deoxy-6-yl-4-O-(2′-naphthyl)methyl-1-thio-α-d-altropyranoside]-1,8-diazabicyclo(5.4.0)undec-7-ene-iodide* (**79**). Compound **73** (257 mg, 0.351 mmol) was converted to **78** according to general **Method B**. The crude product was purified by silica gel chromatography (7:3 CH_2_Cl_2_/n-hexane → 9:1 CH_2_Cl_2_/MeOH) to give **78** (105 mg, 50%) as a white foam and **79** (130 mg, 42%) as a light yellow syrup.

Data of **79**: [α]_D_^25^ + 75.0 (*c* 0.16, CHCl_3_); *R*_f_ 0.54 (9:1 CH_2_Cl_2_/MeOH); ^1^H NMR (500 MHz, CDCl_3_) *δ* = 8.31–7.27 (m, 22H, arom.), 5.96 (s, 1H, H-3), 5.72 (s, 1H, H-1), 5.58 (d, *J* = 3.1 Hz, 1H, H-2), 4.92 (d, *J* = 11.2 Hz, 1H, NAPC*H*_2_a), 4.77 (d, *J* = 11.2 Hz, 1H, NAPC*H*_2_b), 4.71 (t, *J* = 9.6 Hz, 1H, H-5), 4.16 (d, *J* = 15.5 Hz, 1H, H-6a), 3.99 (dd, *J* = 2.5 Hz, *J* = 9.8 Hz, 1H, H-4), 3.76 (dd, *J* = 9.7 Hz, *J* = 15.7 Hz, 1H, H-6b), 3.67–3.49 (m, 6H, 3 × NC*H*_2_ DBU), 2.87–2.74 (m, 2H, C*H*_2_ DBU), 1.92–1.83 (m, 2H, C*H*_2_ DBU), 1.66–1.56 (m, 6H, 3 × C*H*_2_ DBU) ppm; ^13^C NMR (125 MHz, CDCl_3_), *δ* = 167.5 (1C, C_q_ DBU), 165.0, 164.7 (2C, 2 × C_q_ Bz), 134.3, 133.6, 133.0, 132.9, 128.8, 128.3 (6C, 6 × C_q_ arom.), 133.8–126.2 (22C, arom.), 84.6 (1C, C-1), 71.6 (1C, C-4), 71.2 (1C, C-2), 71.1 (1C, NAP*C*H_2_), 67.3 (1C, C-5), 64.9 (1C, C-3), 55.2 (1C, C-6), 55.7, 49.4, 48.4 (3C, 3 × N*C*H_2_ DBU), 29.1, 28.2, 25.7, 22.7, 19.8 (5C, 5 × C*H*_2_ DBU) ppm; UHR ESI-QTOF: *m/z* calcd for C_46_H_47_N_2_O_6_S [M]^+^ 755.3149; found: 755.3154.

*Phenyl 2,3-di-O-benzoyl-4-O-(2′-naphthyl)methyl-1-thio-α-d-arabino-hex-5-enopyranoside* (**78**) and *1,5-anhydro-2-O-benzoyl-3,6-dideoxy-4-O-(2′-naphthyl)methyl-3-S-phenyl-3-thio-α-d-erythro-hex-1,5-dienytol* (**47**). Compound **73** (234 mg, 0.320 mmol) was converted to **78** according to general **Method F**. The crude product was purified by silica gel chromatography (7:3 CH_2_Cl_2_/n-hexane) to give **78** (103 mg, 53%) as a white foam and **47** (20 mg, 10%) as a colorless syrup.

*Phenyl 2,3-di-O-benzyl-4-O-(2′-naphthyl)methyl-1-thio-β-l-galactopyranoside* (**80**). Compound **74** (143 mg, 0.249 mmol) was converted to **80** according to general **Method C**. The crude product was purified by silica gel chromatography (55:45 *n*-hexane/EtOAc) to give **80** (100 mg, 68%) as a colorless syrup. [α]_D_^25^ + 7.0 (*c* 0.10, CHCl_3_); *R*_f_ 0.41 (55:45 *n*-hexane/EtOAc); ^1^H NMR (400 MHz, CDCl_3_) *δ* = 7.84–7.17 (m, 22H, arom.), 5.12–4.65 (m, 6H, NAPC*H*_2_, 2 × BnC*H*_2_), 4.66 (d, *J* = 9.6 Hz, 1H, H-1), 3.99 (t, *J* = 9.4 Hz, 1H, H-2), 3.88–3.84 (m, 2H, H-4, H-6a), 3.61 (dd, *J* = 2.7 Hz, *J* = 9.2 Hz, 1H, H-3), 3.53–3.49 (m, 1H, H-6b), 3.45–3.42 (m, 1H, H-5), 1.84 (d, *J* = 4.7 Hz, 1H, C-6-O*H*) ppm; ^13^C NMR (100 MHz, CDCl_3_) *δ* = 138.3, 138.2, 135.8, 134.1, 133.2, 133.1 (6C, 6 × C_q_ arom.), 131.6–126.2 (22C, arom.), 87.8 (1C, C-1), 84.4 (1C, C-3), 79.0 (1C, C-5), 77.6 (1C, C-2), 75.8, 74.4, 73.2 (3C, NAP*C*H_2_, 2 × Bn*C*H_2_), 73.4 (1C, C-4), 62.4 (1C, C-6) ppm; ESI-TOF-MS: *m/z* calcd for: C_37_H_36_NaO_5_S [M + Na]^+^ 615.2176; found: 615.2189.

*Phenyl 2,3-di-O-benzoyl-4-O-(2′-naphthyl)methyl-1-thio-β-l-galactopyranoside* (**81**). Compound **78** (193 mg, 0.319 mmol) was converted to **81** according to general **Method C**. The crude product was purified by silica gel chromatography (55:45 *n*-hexane/EtOAc) to give **81** (152 mg, 76%) as a colorless syrup. [α]_D_^25^ − 68.3 (*c* 0.12, CHCl_3_); *R*_f_ 0.40 (55:45 *n*-hexane/EtOAc); ^1^H NMR (500 MHz, CDCl_3_) *δ* = 7.98–7.26 (m, 22H, arom.), 5.93 (t, *J* = 9.9 Hz, 1H, H-2), 5.38 (d, *J* = 9.9 Hz, 1H, H-3), 4.97 (d, *J* = 9.9 Hz, 1H, H-1), 4.90 (d, *J* = 11.8 Hz, 1H, NAPC*H*_2_a), 4.68 (d, *J* = 11.8 Hz, 1H, NAPC*H*_2_b), 4.24 (s, 1H, H-4), 3.98–3.95 (m, 1H, H-6a), 3.81 (t, *J* = 5.6 Hz, 1H, H-5), 3.65–3.63 (m, 1H, H-6b), 1.73 (d, *J* = 6.0 Hz, 1H, C-6-O*H*) ppm; ^13^C NMR (125 MHz, CDCl_3_) *δ* = 166.0, 165.4 (2C, 2 × C_q_ Bz), 134.9, 133.2, 133.1, 132.8, 129.7, 129.0 (6C, 6 × C_q_ arom.), 133.6–126.1 (22C, arom.), 86.7 (1C, C-1), 79.3 (1C, C-5), 76.1 (1C, C-3), 75.0 (1C, NAP*C*H_2_), 74.0 (1C, C-4), 68.6 (1C, C-2), 62.1 (1C, C-6) ppm; ESI-TOF-MS: *m/z* calcd for: C_37_H_32_NaO_7_S [M + Na]^+^ 643.1761; found: 643.1768.

*Phenyl 2,3-di-O-benzyl-4,6-O-(2′-naphthyl)methylidene-1-thio-β-l-galactopyranoside* (**82**). Compound **80** (85 mg, 0.144 mmol) was converted to **82** according to general **Method G**. The crude product was purified by silica gel chromatography (CH_2_Cl_2_) to give **82** (68 mg, 80%) as a colorless syrup. [α]_D_^25^ − 30.0 (*c* 0.12, CHCl_3_); *R*_f_ 0.47 (6:4 *n*-hexane/EtOAc); ^1^H NMR (500 MHz, CDCl_3_) *δ* = 7.98–7.14 (m, 22H, arom.), 5.61 (s, 1H, H_ac_), 4.73–4.69 (m, 4H, 2 × BnC*H*_2_), 4.61 (d, *J* = 9.6 Hz, 1H, H-1), 4.37 (dd, *J* = 1.2 Hz, *J* = 12.3 Hz, 1H, H-6a), 4.16 (d, *J* = 3.3 Hz, 1H, H-4), 3.97 (dd, *J* = 1.3 Hz, *J* = 12.3 Hz, 1H, H-6b), 3.93 (t, *J* = 9.4 Hz, 1H, H-2), 3.62 (dd, *J* = 3.4 Hz, *J* = 9.2 Hz, 1H, H-3), 3.35 (s, 1H, H-5) ppm; ^13^C NMR (125 MHz, CDCl_3_) *δ* = 138.6, 138.3, 135.4, 133.9, 133.0, 132.9 (6C, 6 × C_q_ arom.), 132.8–124.4 (22C, arom.), 101.5 (1C, C_ac_), 86.7 (1C, C-1), 81.5 (1C, C-3), 75.3 (2C, C-2, Bn*C*H_2_), 73.9 (1C, C-4), 71.9 (1C, Bn*C*H_2_), 70.0 (1C, C-5), 69.6 (1C, C-6) ppm; ESI-TOF-MS: *m/z* calcd for: C_37_H_34_NaO_5_S [M + Na]^+^ 613.2019; found: 613.2054.

*Phenyl 2,3-di-O-benzoyl-4,6-O-(2′-naphthyl)methylidene-1-thio-β-l-galactopyranoside* (**83**). Compound **81** (149 mg, 0.239 mmol) was converted to **83** according to general **Method G**. The crude product was purified by silica gel chromatography (6:4 *n*-hexane/EtOAc) to give **83** (115 mg, 78%) as a white crystal. [α]_D_^25^ − 111.0 (*c* 0.10, CHCl_3_); M.p.: 226–228 °C (EtOAc/*n*-hexane); *R*_f_ 0.44 (6:4 *n*-hexane/EtOAc); ^1^H NMR (400 MHz, CDCl_3_) *δ* = 7.99–7.19 (m, 22H, arom.), 5.85 (t, *J* = 9.9 Hz, 1H, H-2), 5.61 (s, 1H, H_ac_), 5.40 (dd, *J* = 3.4 Hz, *J* = 9.9 Hz, 1H, H-3), 4.94 (d, *J* = 9.8 Hz, 1H, H-1), 4.61 (d, *J* = 3.2 Hz, 1H, H-4), 4.43 (dd, *J* = 0.9 Hz, *J* = 12.2 Hz, 1H, H-6a), 4.06 (dd, *J* = 1.1 Hz, *J* = 12.3 Hz, 1H, H-6b), 3.66 (s, 1H, H-5) ppm; ^13^C NMR (100 MHz, CDCl_3_) *δ* = 166.3, 165.0 (2C, 2 × C_q_ Bz), 135.1, 133.7, 132.8, 131.2, 129.7, 129.1 (6C, 6 × C_q_ arom.), 133.8–124.2 (22C, arom.), 101.2 (1C, C_ac_), 85.3 (1C, C-1), 74.1 (1C, C-3), 73.8 (1C, C-4), 69.9 (1C, C-5), 69.2 (1C, C-6), 67.2 (1C, C-2) ppm; ESI-TOF-MS: *m/z* calcd for: C_37_H_30_NaO_7_S [M + Na]^+^ 641.1604; found: 641.1616.

*Phenyl 2,3-di-O-benzyl-6-O-(2′-naphthyl)methyl-1-thio-β-l-galactopyranoside* (**84**). Compound **82** (59 mg, 0.099 mmol) was converted to **84** according to general **Method H**. The crude product was purified by silica gel chromatography (98:2 CH_2_Cl_2_/acetone) to give **84** (40 mg, 68%) as a colorless syrup. [α]_D_^25^ + 8.0 (*c* 0.15, CHCl_3_); *R*_f_ 0.35 (98:2 CH_2_Cl_2_/acetone); ^1^H NMR (400 MHz, CDCl_3_) *δ* = 7.84–7.20 (m, 22H, arom.), 4.84–4.66 (m, 6H, NAPC*H*_2_, 2 × BnC*H*_2_), 4.65 (d, *J* = 9.8 Hz, 1H, H-1), 4.11 (d, *J* = 1.7 Hz, 1H, H-4), 3.87–3.79 (m, 2H, H-6a,b), 3.76 (t, *J* = 9.3 Hz, 1H, H-2), 3.63 (t, *J* = 5.7 Hz, 1H, H-5), 3.58 (dd, *J* = 3.2 Hz, *J* = 8.9 Hz, 1H, H-3), 2.56 (s, 1H, C-4-O*H*) ppm; ^13^C NMR (100 MHz, CDCl_3_) *δ* = 138.3, 137.8, 135.5, 134.0, 133.4, 133.1 (6C, 6 × C_q_ arom.), 131.9–125.9 (22C, arom.), 87.8 (1C, C-1), 82.7 (1C, C-3), 77.2 (1C, C-5), 77.1 (1C, C-2), 75.9, 74.0, 72.3 (3C, NAP*C*H_2_, 2 × Bn*C*H_2_), 69.9 (1C, C-6), 67.1 (1C, C-4) ppm; ESI-TOF-MS: *m/z* calcd for: C_37_H_36_NaO_5_S [M + Na]^+^ 615.2176; found: 615.2161.

*Phenyl 2,3-di-O-benzoyl-6-O-(2′-naphthyl)methyl-1-thio-β-l-galactopyranoside* (**85**). Compound **83** (107 mg, 0.173 mmol) was converted to **85** according to general **Method H**. The crude product was purified by silica gel chromatography (98:2 CH_2_Cl_2_/acetone) to give **85** (68 mg, 64%) as a white foam. [α]_D_^25^ −7 7.0 (*c* 0.10, CHCl_3_); *R*_f_ 0.32 (98:2 CH_2_Cl_2_/acetone); ^1^H NMR (400 MHz, CDCl_3_) *δ* = 7.98–7.19 (m, 22H, arom.), 5.82 (t, *J* = 9.9 Hz, 1H, H-2), 5.34 (dd, *J* = 2.9 Hz, *J* = 9.8 Hz, 1H, H-3), 4.96 (d, *J* = 10.0 Hz, 1H, H-1), 4.74 (s, 2H, NAPC*H*_2_), 4.43 (s, 1H, H-4), 3.94–3.92 (m, 1H, H-5), 3.89–3.87 (m, 2H, H-6a,b), 2.86 (s, 1H, C-4-O*H*) ppm; ^13^C NMR (100 MHz, CDCl_3_) *δ* = 166.0, 165.4 (2C, 2 × C_q_ Bz), 135.2, 133.4, 133.2, 132.7, 129.6, 129.2 (6C, 6 × C_q_ arom.), 133.5–125.8 (22C, arom.), 86.7 (1C, C-1), 77.4 (1C, C-5), 75.7 (1C, C-3), 74.0 (1C, NAP*C*H_2_), 69.7 (1C, C-6), 68.4 (1C, C-4), 68.1 (1C, C-2) ppm; ESI-TOF-MS: *m/z* calcd for: C_37_H_32_NaO_7_S [M + Na]^+^ 643.1761; found: 643.1779.

*Phenyl 2,3-di-O-benzyl-6-O-(2′-naphthyl)methyl-4-O-(4′-nitrobenzoyl)-1-thio-β-l-glucopyranoside* (**86**). Compound **84** (24 mg, 0.040 mmol) was converted to **86** according to general **Method K**. The crude product was purified by silica gel chromatography (7:3 *n*-hexane/EtOAc) to give **86** (18 mg, 60%) as a colorless syrup. [α]_D_^25^ + 12.0 (*c* 0.10, CHCl_3_); *R*_f_ 0.63 (7:3 *n*-hexane/EtOAc); ^1^H NMR (500 MHz, CDCl_3_) *δ* = 7.95–7.03 (m, 26H, arom.), 5.31 (t, *J* = 9.4 Hz, 1H, H-4), 4.95–4.52 (m, 7H, H-1, NAPC*H*_2_, 2 × BnC*H*_2_), 3.80 (t, *J* = 9.1 Hz, 1H, H-3), 3.76–3.73 (m, 1H, H-5), 3.71–3.66 (m, 2H, H-6a,b), 3.64 (t, *J* = 9.3 Hz, 1H, H-2) ppm; ^13^C NMR (125 MHz, CDCl_3_) *δ* = 163.7, 150.2 (2C, 2 × C_q_ *p*-NO_2_-Bz), 137.9, 137.8, 135.0, 134.9, 133.5, 133.0 (6C, 6 × C_q_ arom.), 132.2–123.2 (26C, arom.), 88.0 (1C, C-1), 83.5 (1C, C-3), 81.0 (1C, C-2), 77.0 (1C, C-5), 75.7, 75.5, 73.8 (3C, NAP*C*H_2_, 2 × Bn*C*H_2_), 72.6 (1C, C-4), 70.1 (1C, C-6) ppm; ESI-TOF-MS: *m/z* calcd for: C_44_H_39_NNaO_8_S [M + Na]^+^ 764.2289; found: 764.2139.

*Phenyl 2,3-di-O-benzoyl-6-O-(2′-naphthyl)methyl-4-O-(4′-nitrobenzoyl)-1-thio-β-l-glucopyranoside* (**87**) and *Phenyl 2,3-di-O-benzoyl-6-O-(2′-naphthyl)methyl-4-deoxy-1-thio-α-d-threo-hex-4-enopyranoside* (**88**). Compound **85** (40 mg, 0.063 mmol) was converted to **87** according to general **Method K**. The crude product was purified by silica gel chromatography (CH_2_Cl_2_) to give **87** (25 mg, 51%) as a colorless syrup and **88** (7 mg, 18%) as a colorless syrup.

Data of **87**: [α]_D_^25^ + 20.9 (*c* 0.11, CHCl_3_); *R*_f_ 0.55 (CH_2_Cl_2_); ^1^H NMR (500 MHz, CDCl_3_) *δ* = 7.96–7.26 (m, 26H, arom.), 5.85 (t, *J* = 9.4 Hz, 1H, H-3), 5.56 (t, *J* = 9.7 Hz, 1H, H-4), 5.51 (t, *J* = 9.7 Hz, 1H, H-2), 5.05 (d, *J* = 10.0 Hz, 1H, H-1), 4.64 (s, 2H, NAPC*H*_2_), 4.07–4.05 (m, 1H, H-5), 3.84–3.78 (m, 2H, H-6a,b) ppm; ^13^C NMR (125 MHz, CDCl_3_) *δ* = 165.9, 165.2, 163.7, 150.4 (4C, 2 × C_q_ Bz, 2 × C_q_ *p*-NO_2_-Bz), 134.9, 134.4, 133.0, 132.1, 129.3, 128.8 (6C, 6 × C_q_ arom.), 133.5–123.3 (26C, arom.), 86.6 (1C, C-1), 77.4 (1C, C-5), 74.4 (1C, C-3), 73.9 (1C, NAP*C*H_2_), 71.3 (1C, C-4), 70.6 (1C, C-2), 69.5 (1C, C-6) ppm; ESI-TOF-MS: *m/z* calcd for: C_44_H_35_NNaO_10_S [M + Na]^+^ 792.1874; found: 792.1800.

Data of **88**: [α]_D_^25^ + 49.5 (*c* 0.62, CHCl_3_); *R*_f_ 0.41 (CH_2_Cl_2_); ^1^H NMR (500 MHz, CDCl_3_) *δ* = 8.20–7.26 (m, 22H, arom.), 5.86 (s, 1H, H-1), 5.75 (s, 1H, H-2), 5.54 (s, 2H, H-3, H-4), 4.78 (dd, *J* = 11.8 Hz, *J* = 30.2 Hz, 2H, NAPC*H*_2_), 4.18 (d, *J* = 13.3 Hz, 1H, H-6a), 4.10 (d, *J* = 13.1 Hz, 1H, H-6b) ppm; ^13^C NMR (125 MHz, CDCl_3_) *δ* = 165.6, 165.5 (2C, 2 × C_q_ Bz), 152.0 (1C, C-5), 135.9, 135.3, 134.5, 129.9, 129.2 (6C, 6 × C_q_ arom.), 133.7–126.0 (22C, arom.), 97.3 (1C, C-4), 84.0 (1C, C-1), 72.5 (1C, NAP*C*H_2_), 70.0 (1C, C-2), 69.6 (1C, C-6), 64.9 (1C, C-3) ppm; ESI-TOF-MS: *m/z* calcd for: C_37_H_30_NaO_6_S [M + Na]^+^ 625.1655; found: 625.1632.

## 4. Conclusions

We have successfully developed an efficient synthetic route for the preparation of four rare l-hexoses (l-gulose, l-galactose, l-allose, and l-glucose) in the form of orthogonally protected thioglycosides, starting from the cheap d-mannose, using the most readily available chemicals and most cost-effective transformations possible. The preparation of the 5,6-unsaturated derivatives, required for the C-5 epimerization, was investigated systematically in the presence of ether and ester protecting groups. We have found that the outcome of the elimination reactions is strongly dependent on both the protecting group pattern, the sugar configuration, and the reagent applied. For the fully ether protected derivative, elimination with NaH reagent led to the best yields. AgF-induced dehydrohalogenation proceeded efficiently from ester-bearing compounds, however, in the case of C-3 ester group, a glycal by-product has also been formed due to elimination and subsequent allylic rearrangement. The only example for such an intriguing side reaction has previously been observed in photoinitiated thiol-ene reaction of 2,3-unsaturated α-thioglycosides [38]. In DBU-induced reactions, significant amounts of amidinium salt by-products were formed in all cases, regardless of the type and configuration of the protecting groups. C-5 epimerization could be performed with good stereoselectivity for all derivatives, but the yields were significantly higher for ester-protected derivatives than for ether protected ones. Oxidation-reduction-based C-4 epimerization of vicinal *trans* diols to vicinal *cis* diols was performed with good stereoselectivity in the presence of an ether protecting group adjacent to the keto functionality. The same method yielded vicinal *trans* diol in the presence of an ester group adjacent to the oxo group, which was exploited during the d-mannose to d-altrose conversion. The Mitsunobu inversion proved to be suitable for the preparation of equatorial *trans* diols, but an undesired elimination side reaction was also observed in the presence of an adjacent ester group. All designed l-hexoses were successfully prepared in 9–15 steps (total yields for l-gulose: 21–23%; for l-galactose: 6–8%; for l-allose: 6–8%; for l-glucose: 2–3%) as thioglycosides suitable for the synthesis of oligosaccharides, which can facilitate the synthesis of biologically active molecules.

## Data Availability

Not applicable.

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
