# Peer review of "Synthesis of Four Orthogonally Protected Rare l-Hexose Thioglycosides from d-Mannose by C-5 and C-4 Epimerization"

_molecules, 2022, doi:10.3390/molecules27113422_

Round 1

Reviewer 1 Report

The authors made an extensive study on the production of 4 L-hexose thioglycosides, which greatly expands upon their own previous work as well as on the state of the art known in literature.

While many of the reaction processes applied by the authors are known by literature and "only" applied to new substrates, which may in itself arguably not be enough for the publication in Molecules, the high scientific significance of this work, in my opinion, stems greatly from the extensive work done by the authors to investigate the elimination processes towards the intermediates with an exocyclic double bond with DBU, NaH and AgF. This allowed the authors to find the best reaction procedure towards each individual L-hexose thioglycoside.

All explanations and discussions done by the authors seem sound and their supplementary data containing the NMR shows an overall high purity of their substrates.

I recommend the acceptance of this manuscript in the presented form.

Author Response

I would sincerely like to thank the reviewer for constructive comments which helped us in improving the manuscript.

Reviewer 2 Report

In this study, the authors developed a reaction pathway to obtain two orthogonally protected L-hexose thioglycoside derivatives, L-gulose and L-galactose, through the corresponding 5,6-unsaturated thioglycosides by C-5 epimerization. The authors further synthesized the orthogonally protected thioglycosides of L-allose and L-glucose from these derivatives by C-4 epimerization. This reviewer recommends that it be published after minor revisions.

  1. The authors should give the total yields for their final products in abstract and in conclusion.
  2. For the introduction in page 1, the authors should give some structures of natural products containing L-sugars (ref1 - 8).
  3. For selective benzylation (compound 29 - 30), there are several better methods than the method chosen by the authors and the yields of 30 could be improved (from 73% to 90%, Adv. Synth. Catal. 2014, 356, 1735; Green Chem., 2020, 22, 1139; etc).
  4. For the elimination reaction parts, the authors may try to use TEA or DIPEA as the base, because both TEA and DIPEA should have a greater steric hindrance effect than DBU for N-attacking.

Author Response

  1. “The authors should give the total yields for their final products in abstract and in conclusion.”

Answer: The total yields of the final products requested by the reviewer are included in both the abstract and the conclusion.

  1. “For the introduction in page 1, the authors should give some structures of natural products containing L-sugars (ref1 - 8).”

Answer: The introduction part was supplemented with a figure about some structures of natural products containing L-sugars as it was suggested by the reviewer.

  1. „For selective benzylation (compound 29-30), there are several better methods than the method chosen by the authors and the yields of 30 could be improved (from 73% to 90%, Adv. Synth. Catal. 2014, 356, 1735; Green Chem., 2020, 22, 1139; etc).”

Answer: Thanks for the suggestion, by reviewing the literature provided, we will try the suggested methods in the future.

  1. „For the elimination reaction parts, the authors may try to use TEA or DIPEA as the base, because both TEA and DIPEA should have a greater steric hindrance effect than DBU for N-attacking.”

Answer: Thank you for your suggestion. For other compounds (d-allose derivatives) we have already tried to induce the elimination with other bases (ttBP, K2CO3, DABCO and TEA), unfortunately none of the bases was suitable, the expected products were not formed. DIPEA has not been tested yet, but we will definitely investigate it in the future to induce elimination.

Reviewer 3 Report

In this manuscript the authors utilized an efficient synthetic route for the preparation four rare L-hexoses in the form of orthogonally protected thioglycosides via C4/C5 epimerization using elimination/hydroboration and oxidation. Thus, the overall amount of novelty is not remarkably high as well as quality of the Supporting information is not reasonable. It is,
however, in my opinion, ample to allow publication, after addressing a
few issues:

Concerns to authors

  • There are alert levels in the author generated check CIF report that need to be addressed. The CIF should be examined and corrected by authors. CCDC NO of the compound should be included in the text.
  • NMR spectra of many compounds look like some kind of DEPT but no single information about that. Is it DEPT45, DEPT90 or DEPT135? In addition, 13C NMR spectra in the experimental part are not reported as DEPT measurements.
  • - Number of 1H and 13C NMR signals in SI not always match the molecular structures. It seems that some impurities and artifacts are marked but this must be clearly explained. I strongly recommend check carefully the whole experimental part and SI. Now it looks quite messy.
  • COSY/HSQC correlation should be specified at least few of the compounds.
  • Spacing is missing in many places along the whole manuscript, e.g. between value and unit. Weights should be uniform in all cases--Line547, 554, 584
  • Compounds numbering carefully checked for example, compound 28 did not see in the discussion.

Author Response

  1. “There are alert levels in the author generated check CIF report that need to be addressed. The CIF should be examined and corrected by authors. CCDC NO of the compound should be included in the text.”

Answer: Thank you for your remark. The paper is augmented with the depository information. Moreover, the attached checkcif and cif files contain the Validation Reply Form with explanation for A and B level errors. The correctness of the structure is not influenced by these errors.

  1. “NMR spectra of many compounds look like some kind of DEPT but no single information about that. Is it DEPT45, DEPT90 or DEPT135? In addition, 13C NMR spectra in the experimental part are not reported as DEPT measurements.”

Answer: The spectra given in SI are not DEPT measurements but J-modulated 13C measurements. We chose this measurement method because more information can be extracted from it than from DEPT measurements. For example, quaternary carbon atoms can be measured. In the spectra shown in SI, the upward signals may be -CH-s or -CH3-s, and the downward signals may be -CH2-s or quaternary carbons.

  1. “Number of 1H and 13C NMR signals in SI not always match the molecular structures. It seems that some impurities and artifacts are marked but this must be clearly explained. I strongly recommend check carefully the whole experimental part and SI. Now it looks quite messy.”

Answer: We were checked the NMR spectra and the SI and no discrepancies were found. However, for some compounds 1H spectra contains a small amount of solvents (e. g. EtOAc or n-hexane) and some grease. Unfortunately, since these compounds are formed as syrups, they are not always sufficiently dried and may contain solvent residues and may also show signs of impurities remaining in solvents of poor purity. In the future, we will put more emphasis on eliminating these problems.

  1. “COSY/HSQC correlation should be specified at least few of the compounds.”

Answer: 64 previously unknown new compounds have been described in this publication. 1H, 13C (J-modulated), COSY and HSQC NMR spectra of most of these compounds (61) are shown in the SI.

  1. “Spacing is missing in many places along the whole manuscript, e.g. between value and unit. Weights should be uniform in all cases--Line547, 554, 584”

Answer: Spaces were checked and corrected where necessary. Furthermore, the weights were standardized in the experimental section.

  1. “Compounds numbering carefully checked for example, compound 28 did not see in the discussion.”

Answer: The numbering of the compounds in the article was checked and, where necessary, corrected (line 295: 77 to 78). The number of compound 28 was indeed not included in the text, this has been corrected.

Round 2

Reviewer 3 Report

I am ok with the improvements made by the Authors and recommend it to published.